# Estimating the potential impact and diagnostic requirements for SARS-CoV-2 test-and-treat programs

Alvin X. Han [1] ✉, Emma Hannay[2], Sergio Carmona[2], Bill Rodriguez[2], Brooke E. Nichols[1,2,3,4] & Colin A. Russell [1,3,4] ✉

Oral antivirals have the potential to reduce the public health burden of COVID-19. However, now that we have exited the emergency-phase of the COVID-19 pandemic, declining SARS-CoV-2 clinical testing rates (average testing rates = ≪10 tests/100,000 people/day in low-and-middle income countries; <100 tests/100,000 people/day in high-income countries; September 2023) make the development of effective test-and-treat programs challenging. We used an agent-based model to investigate how testing rates and strategies affect the use and effectiveness of oral antiviral test-to-treat programs in four country archetypes of different income levels and demographies. We find that in the post-emergency-phase of the pandemic, in countries where low testing rates are driven by limited testing capacity, significant population-level impact of test-and-treat programs can only be achieved by both increasing testing rates and prioritizing individuals with greater risk of severe disease. However, for all countries, significant reductions in severe cases with antivirals are only possible if testing rates were substantially increased with high willingness of people to seek testing. Comparing the potential population-level reductions in severe disease outcomes of test-to-treat programs and vaccination shows that test-and-treat strategies are likely substantially more resource intensive requiring very high levels of testing (≫100 tests/100,000 people/day) and antiviral use suggesting that vaccination should be a higher priority.

Antiviral therapies such as anti-SARS-CoV-2 monoclonal antibodies, replication inhibitors, protease inhibitors, and host-directed therapies can be used to treat COVID-19, reducing the probability of severe disease to varying degrees[1]. Direct-acting antiviral drugs, such as molnupiravir[2] and nirmatrelvir–ritonavir (Paxlovid)[3], have the potential to substantially lower disease burden given their efficacy and convenience of oral dosing. Nirmatrelvir–ritonavir, in particular, can reduce incidence of adverse events in high-risk individuals (i.e., ≥60 years of age (over-60 y) or an adult ≥18 years with a relevant comorbidity) by 46–89%[3,4]. Given their ability to lower viral load[3],

these drugs could also potentially be used to control SARS-CoV-2 transmission[5]. To achieve maximum impact, these drugs must typically be administered within a few days of symptom onset. Given limited resources and the relatively high cost of these drugs[6], along with the need to administer drugs quickly after symptom onset[2,3], diagnostic testing remains an essential first step for identifying suitable drug recipients.

Oral antivirals (the term "antivirals" refers only to oral direct antivirals for the rest of this article) have the potential to reduce the disease burden of COVID-19 outbreaks. Various studies have

[1]Department of Medical Microbiology & Infection Prevention, Amsterdam University Medical Center, University of Amsterdam, Amsterdam, The Netherlands. [2]Foundation for Innovative New Diagnostics (FIND), Geneva, Switzerland. [3]Department of Global Health, School of Public Health, Boston University, Boston, MA, USA. [4]These authors contributed equally: Brooke E. Nichols, Colin A. Russell. ✉e-mail: x.han@amsterdamumc.nl; c.a.russell@amsterdamumc.nl

estimated ~10–40% reduction in severe disease outcomes if antivirals were distributed to 20–50% of all symptomatic infected individuals[5,7,8]. However, none of these studies have accounted for the diagnostic capacity required to identify and treat these cases with antivirals. There have been substantial gaps in COVID-19 testing equity across country income groups throughout the pandemic. Between January 2020 and March 2022, LMICs were only testing at an average of 27 tests/100,000 people/day (tests/100 K/day) as compared to >800 tests/100 K/day in high-income countries (HICs)[9]. In the post-public health emergency phase of the pandemic, testing rates have dwindled down to less than 10 tests/100 K/day and 100 tests/100 K/day on average for LMICs and HICs respectively (as of September 2023)[9]. Low testing rates severely underestimate COVID-19 cases[10], which not only complicate antiviral demand forecasts but also create additional barriers to the effective distribution and use of antivirals.

Here, we used an agent-based model (PATAT)[11,12] to demonstrate how testing rates and strategies affect the use and impact of antivirals. In the model, we focused on antigen rapid diagnostic tests (Ag-RDTs) which can easily be performed at point-of-care or be used as self-tests with short turnaround time needed to quickly identify high-risk infected individuals[13]. We computed the potential impact of test-and-treat programs on infections, severe cases, and deaths averted in three LMIC archetypes with distinct demographic structures—Brazil, Georgia, and Zambia—and the Netherlands as an HIC example, all under varying levels of vaccination coverage. The LMIC archetypes were selected as the age demography of their populations were largely representative of the 132 other LMICs as classified by the World Bank (Fig. 1)[14,15]. Our findings highlight the limits and expected outcomes of COVID-19 oral antiviral treatment programs under realistic testing and vaccination landscapes.

## Results

### Dynamic epidemic simulations with PATAT

We first provide key details of the PATAT model and assumptions to contextualize our results. See Methods and Supplementary Information for full description of the model and parameters. We simulated SARS-CoV-2 epidemics in each country under a range of average effective reproduction number (i.e., $R_e$ = 0.9, 1.2 (doubling time = 6–9 days), 1.5 (doubling time = 3–5 days), and 2.0 (doubling time = 1–3 days)) during the first week of each simulation. These doubling times coincide the range reported for prominent Omicron subvariants as well, including BA.2 (~3 days)[16], BA.5 (5–6 days)[17] and XBB.1.5 (9–10 days)[18] (Figure S1). All simulations were initialized with 1% of the population infected at the start of the epidemic. We did not model varying levels of population immunity due to the lack of comprehensive country-specific infection data and complexities in parameterizing the proportion and protection conferred to individuals with infections by different variant infection histories in the past. Instead, the different $R_e$ values should be viewed as the collective outcome of population immunity from previous infections, intrinsic transmissibility of the variant virus as well as effects of any existing any public health interventions other than vaccination and oral antivirals. For each $R_e$ value and country, we performed two sets of simulations—one with and the other without the distribution of antivirals. For each set of simulations, we assumed three vaccination coverage: 10%, 50% and 90%. We randomly assigned vaccination status across the simulated population but assumed that vaccination was age-tiered such that the older individuals were vaccinated first. For comparability between countries and as a simplification, we assumed that protection rates against infection and severe disease were 29% and 70%, respectively, which were based on the more conservative, lower average estimates of vaccine effectiveness

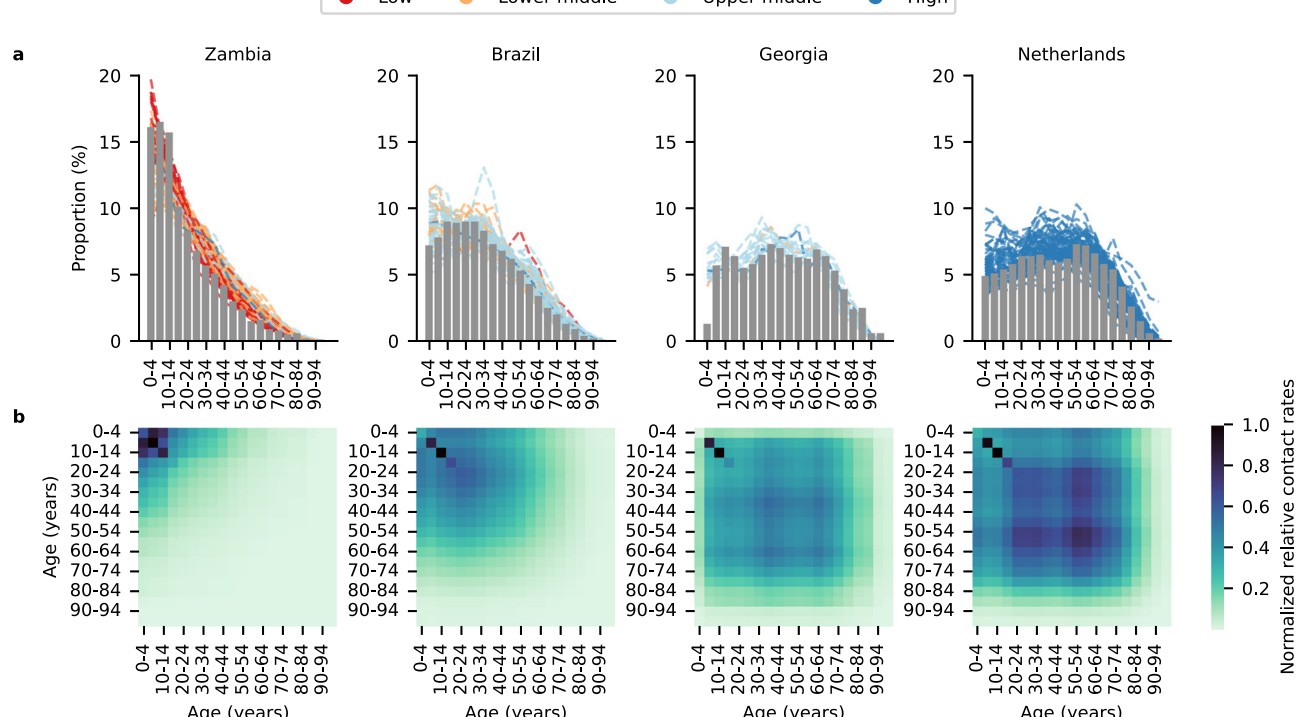

**Fig. 1 | Demography of simulated countries. a** Bar plot shows the age distribution of each simulated country archetype (low and middle-income countries (LMICs): Brazil, Georgia, Zambia; high-income country: Netherlands) stratified in 5-year bins. Each dashed black line in the Brazil, Georgia and Zambia plots denotes the age distribution of one of 132 other LMICs[14] that best matches (i.e., lowest mean absolute error) the age distribution of the simulated country archetype. Age distribution of the population in each country is downloaded from World Population Prospects compiled by the United Nations[15]. **b** Heatmap showing the normalized relative contact rates between individuals of different age groups in 5-year bins averaged across all contact networks generated by the PATAT simulation model.

against BA.1 across different vaccines (i.e., mRNA and ChAdOx1 nCoV-19 vaccine) and doses (i.e., 1–3 doses)[19–21].

The relative susceptibility of individuals to infection[22,23], probability of becoming symptomatic[24,25], probability of developing severe disease[24,25], and the probability of death[26,27] depend on the age of the individual (Table S1). We assumed that only high-risk individuals (i.e., ≥60 years of age (over-60y) or an adult ≥18 years with a relevant comorbidity) who tested positive at clinics (e.g., a self-reported self-test would be insufficient to access antivirals) would receive a course of antivirals. We also randomly assigned 20% of the population to have a 40% increase in relative risk to developing severe disease because of pre-existing comorbidities (e.g., obesity, diabetes, people living with HIV, etc.)[28,29]. As a simplification, we assumed that the prevalence of comorbidities was independent of age. Although the phase 2/3 trial of nirmatrelvir–ritonavir reported 89% relative risk reduction among unvaccinated high-risk patients infected by the Delta variant-of-concern[3], we assumed that an antiviral course conferred a 46% risk reduction for infected high-risk individuals to severe disease outcomes based on a separate cohort study on the effectiveness of nirmatrelvir–ritonavir among high-risk patients infected by Omicron BA.1 independent of their vaccination status[4].

## Impact of test-and-treat
We simulated the implementation of test-and-treat programs during SARS-CoV-2 epidemic waves in three different LMICs (Brazil, Georgia, and Zambia) with distinct population demography (Fig. 1) and the Netherlands under different levels of vaccine coverage (10%, 50% or 90%) and average test availability (10, 100, or 500 tests/100 K/day). We assumed that tests were only available at health clinics and that 65% of individuals with mild symptoms would likely seek testing at clinics based on surveys of testing behavior during the pandemic[30,31]. Test-seeking individuals would, however, only be tested if tests were available. From our simulations, we found that the likelihood of detecting an infection ranged between 0.06% and 64.6%, depending on the country simulated, epidemic intensity, vaccination coverage and test availability (Figure S2). Generally, detection is more likely with a larger proportion of over-60y individuals (i.e., the more likely cases will be symptomatic and seek testing), lower reproduction rate $R_e$, higher vaccination coverage and greater test availability (i.e., any of the aforementioned factors directly or indirectly increases the surplus of tests available for symptomatic individuals).

At 10 tests/100 K/day, test-and-treat programs are unlikely to have any population-level impact on disease transmission in all countries (Figure S3). At higher testing rates (≥100 tests/100 K/day) and lower $R_e$ (≤1.5), there were modest differences between simulated countries. We found that antivirals largely only have a limited impact on total infections averted (Figure S3), in large part because 58–67% of all transmission events were attributed to asymptomatic and pre-symptomatic individuals (Figure S4A). However, in Georgia and the Netherlands where >30% of the population are over-60y and high-risk individuals transmitted almost half of all infections (Figure S4B), increasing testing rates to 100 (500) tests/100 K/day, accompanied by uncapped distribution of antivirals, could reduce total infections by -12% (-22–24%). On the other hand, regardless of testing rates, infections averted were <12% and <4% in Brazil and Zambia respectively, both of which have smaller over-60y populations (i.e., Brazil: 15%; Zambia: 6% of population; Figure S3A) and where most infections are transmitted by low-risk individuals (Figure S4B). Across all settings and testing rates, increasing vaccination coverage did not change the proportion of infections averted by antivirals substantially.

If testing rates increased to 500 tests/100 K/day, the proportion of severe cases averted due to antivirals would depend on the proportion of over-60y in the population, with Zambia, Brazil, Georgia and the Netherlands, maximally reducing up to an average of 46%, 55%, 67% and 68% of severe cases respectively through test-and-treat strategies

(Fig. 2). Linking antiviral treatment to testing programs at a rate of 10 tests/100 K/day did not generate any impact under any scenario, including when 90% of the population were vaccinated. Raising testing rates to 100 tests/100 K/day—a widely publicized global target during the pandemic—and treating all high-risk, test-positive patients with antivirals substantially increased the proportion of severe cases averted at lower $R_e$ (i.e., proportion of severe cases averted at $R_e$ = 0.9 (1.2) with 10–90% vaccination coverage: Zambia, 17–20% (3–4%); Brazil, 24–55% (6–14%); Georgia, 50–65% (13–30%) and the Netherlands, 48–67% (12–31%); Fig. 2). The impact was greatest in Georgia and the Netherlands given their substantial >60 y population. As $R_e$ increases (≥1.5), the likely population demand for tests also increased, and correspondingly >100 tests/100 K/day was needed to ensure that high-risk individuals could be identified to initiate treatment (i.e., proportion of severe cases averted at $R_e$ = 1.5 (2.0) with 10–90% vaccination coverage at 100 tests/100 K/day: Zambia, 2–4% (0–3%); Brazil, 1–4% (0–1%); Georgia, 3–9% (1–2%); Netherlands, 3–10% (0–2%). At 500 tests/100 K/day: Zambia, 9–16% (7–9%); Brazil, 11–36% (6–9%); Georgia, 24–66% (8–14%); Netherlands, 28–65% (6–18%); Fig. 2). Although we did not model the impact of antivirals in reducing the likelihood of death, developing severe disease precedes dying from COVID-19 in our model (see Methods), the number of deaths averted thus follow similar trends as severe cases averted (Figure S5).

At testing rates of ≤10 tests/100,000 people/day, use of antivirals made negligible contributions to reducing severe disease at all levels of vaccine coverage (Table 1). At testing rates ≥100 tests/100,000/people/day, higher vaccination coverage was associated with a smaller absolute number of severe cases averted by antivirals. However, at higher testing rates, the proportion of severe cases averted by antivirals relative to no distribution of antivirals is larger at higher vaccination coverage. This is because as infections decrease with higher vaccination coverage, a greater percentage of severe cases could also be detected and treated by antivirals assuming that the quantity of test availability is a constraining factor and that demand in low vaccination scenarios would exceed supply.

## Distribution of test and antivirals to high-risk household contacts of test-positive individuals
As antivirals must be administered quickly after symptom onset, one way to promptly identify and treat infected high-risk individuals is to secondarily distribute self-tests to high-risk household contacts who were exposed to the test-positive individuals. This would, however, also result in a faster depletion of available test stocks under limited test availability. We repeated our simulations with high-risk household contacts receiving Ag-RDTs from clinics to perform self-test over the ensuing three days, initiating antiviral treatment upon a positive diagnosis. In this scenario, however, there was little reduction in total infections due to antivirals (Figure S6). In fact, when $R_e$ was low (≤1.2) and at 100 tests/100 K/day, distributing tests to high-risk household contacts for self-tests diverted away test stocks that would otherwise be used to diagnose test-seeking symptomatic individuals (which would, in turn, change their behavior to reduce transmission if tested positive). At 100 tests/100 K/day across all $R_e$ values, or at 500 tests/100 K/day and higher $R_e$, the proportion of severe cases and in turn, deaths averted diminished substantially by a factor of two- to ten-fold (Figures S7–8) relative to no secondary distribution of Ag-RDTs to high-risk household contacts (Fig. 2 and Figure S5). Unless testing rates were increased to ≥500 tests/100 K/day, 100 tests/100 K/day remains inadequate to meet the testing demand of both symptomatic individuals and high-risk household contacts to derive greater impact from test-and-treat.

## Restricting symptomatic testing to high-risk individuals
Given the modest impact of antivirals in reducing transmissions, testing could be targeted to high-risk individuals only in order to

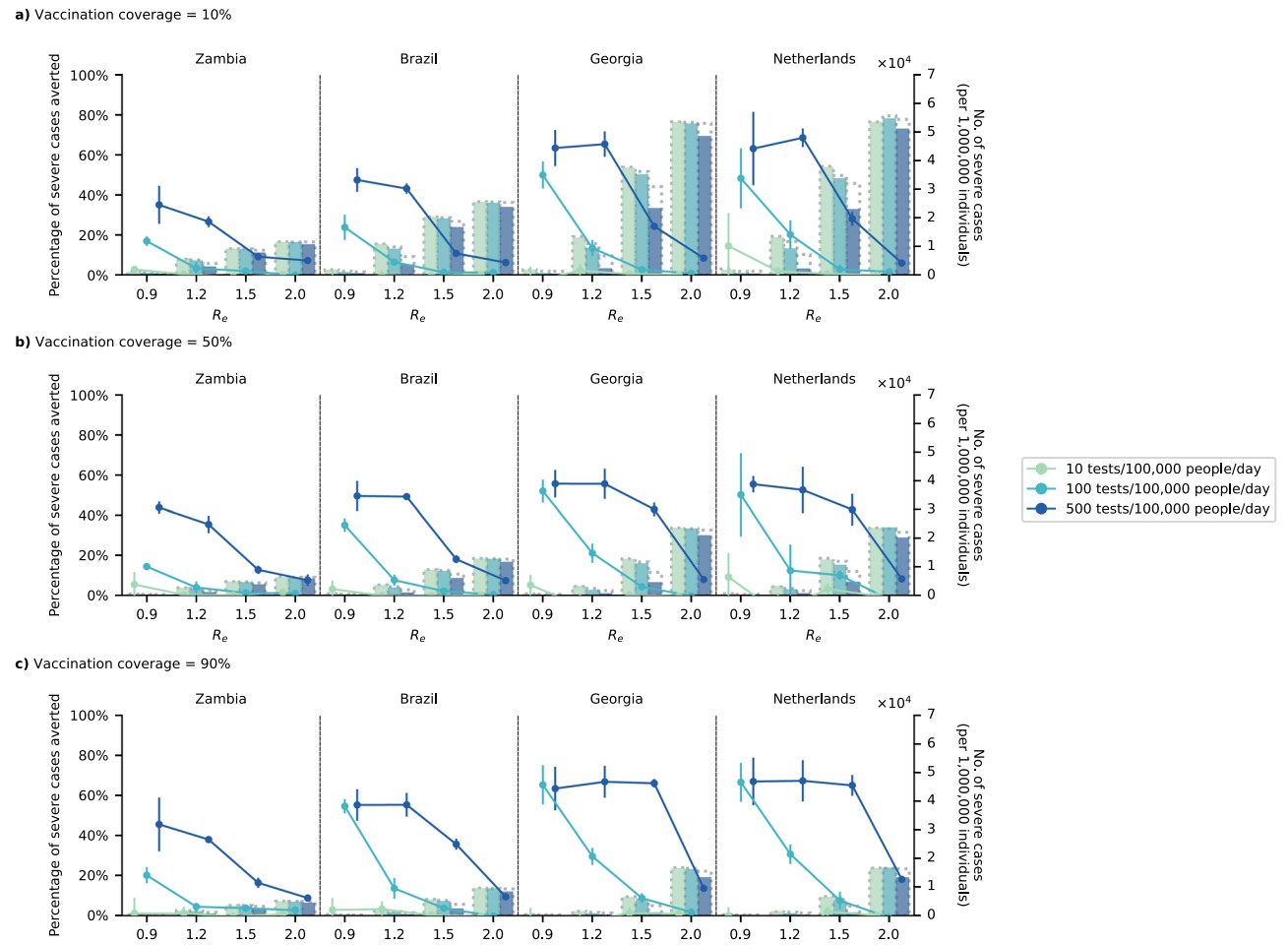

**Fig. 2 | Impact of test-and-treat on severe cases.** No restrictions on access to symptomatic testing at clinics (i.e., all symptomatic individuals who sought testing at clinics would receive one if in stock) and high-risk household contacts of test-positive individuals are not tested. All eligible high-risk individuals (i.e., ≥60 years of age or an adult ≥18 years with a relevant comorbidity) who tested positive were given a course of oral antivirals. Line plots (left axis) show the mean percentage change (standard deviation denoted by error bars; $n = 5$ independent simulations) in severe cases relative to no distribution of antivirals under different levels of mean test availability (different shades of color) after a 90-day epidemic wave in a population of 1,000,000 individuals with **a** 10%, **b** 50%, and **c** 90% vaccination coverage for different epidemic intensities (measured by the initial effective reproduction number ($R_e$); x axis). Bar plots (right y axis) show the number of severe cases in each corresponding scenario. The dotted outline of each bar shows the number of severe cases of each scenario when no antivirals were distributed.

distribute antivirals to as many infected high-risk individuals as possible. This strategy can be effective in reducing severe cases and deaths by test-and-treat when Ag-RDT availability is inadequate to test all symptomatic individuals who seek testing, which was a common scenario in LMICs during the pandemic. Otherwise, if most individuals only isolate themselves after a positive test, the testing restriction would lead to excess tests available that are not effectively used to alter the behavior of low-risk infected individuals that curb onward transmissions.

In our model, restricting testing to high-risk groups when there are ample amount of tests to diagnose non-high-risk symptomatic individuals as well resulted in more transmissions (Figure S9) and severe cases (Fig. 3). We estimated that there can be up to 56% more infections at $R_e \leq 1.5$ if test availability was 500 tests/100 K/day but were restricted to high-risk individuals only. In Georgia, for example, restricting testing to high-risk groups would reduce 52% of severe cases by antivirals at $R_e = 1.5$, 500 tests/100 K/day and 90% vaccination coverage as opposed to 66% under the same scenario but without testing restrictions. On the other hand, when operating under limited test availability relative to $R_e$, restricting symptomatic testing to high-risk individuals could be an effective strategy to

further reduce severe cases (i.e., Fold increase in proportion of severe cases averted relative to no symptomatic testing restrictions when $R_e \geq 1.5$, across all vaccination coverages and countries simulated: 100 tests/100 K/day, median 4.9-fold (IQR = 3.3–6.4); 500 tests/100 K/day, median 3.2-fold (IQR = 2.4–5.1)) and in turn, deaths as well (Figure S10). Of the test distribution strategies simulated in this study, restricting testing to high-risk-groups-only also substantially reduced the number of tests performed per antiviral distributed to median 6 tests (IQR = 5–8 tests; Figure S11). In contrast, a median 20 tests (IQR = 15–33 tests) would be required per antiviral distributed if symptomatic testing was performed without restrictions about risk status.

**Oral antiviral need**

Assuming that only symptomatic high-risk individuals who sought testing received an antiviral course upon a positive test, and that there were two 90-day epidemic waves in a year, we estimated that one antiviral course is needed for every 73–251 (14–154) persons per year on average if testing rate was 100 (500) tests/100 K/day across all simulated countries and vaccination coverage (Fig. 4). We assumed that vaccine protection against infection was low (29%) and that

**Table 1 | Mean number and proportion of severe cases averted due to distribution of oral antivirals at 10% and 90% vaccination coverage**

| Country | Testing rate (tests/ 100,000 people/ day) | $R_e$ | 10% vaccination coverage | | 90% vaccination coverage | |
|---|---|---|---|---|---|---|
| | | | No. of severe cases averted | Proportion of severe cases averted | No. of severe cases averted | Proportion of severe cases averted |
| Zambia | 10 | 0.9 | 29 | 2.7 | 3 | 1.1 |
| | | 1.2 | 16 | 0.3 | 20 | 1.1 |
| | | 1.5 | 0 | 0.0 | 0 | 0.0 |
| | | 2.0 | 22 | 0.2 | 32 | 0.7 |
| | 100 | 0.9 | 147 | 16.9 | 48 | 20.1 |
| | | 1.2 | 170 | 3.3 | 72 | 4.4 |
| | | 1.5 | 172 | 1.9 | 131 | 3.7 |
| | | 2.0 | 0 | 0.0 | 0 | 0.0 |
| | 500 | 0.9 | 242 | 35.0 | 102 | 45.6 |
| | | 1.2 | 1053 | 26.5 | 441 | 38.0 |
| | | 1.5 | 780 | 9.0 | 527 | 16.4 |
| | | 2.0 | 824 | 7.2 | 433 | 8.8 |
| Brazil | 10 | 0.9 | 0 | 0.0 | 10 | 2.8 |
| | | 1.2 | 0 | 0.0 | 59 | 3.2 |
| | | 1.5 | 0 | 0.0 | 13 | 0.2 |
| | | 2.0 | 61 | 0.2 | 0 | 0.0 |
| | 100 | 0.9 | 303 | 23.7 | 133 | 54.6 |
| | | 1.2 | 623 | 6.4 | 195 | 13.6 |
| | | 1.5 | 245 | 1.2 | 196 | 3.7 |
| | | 2.0 | 305 | 1.2 | 24 | 0.2 |
| | 500 | 0.9 | 511 | 47.4 | 134 | 55.3 |
| | | 1.2 | 2739 | 43.1 | 545 | 55.4 |
| | | 1.5 | 2005 | 10.7 | 1404 | 35.7 |
| | | 2.0 | 1553 | 6.1 | 872 | 9.4 |
| Georgia | 10 | 0.9 | 0 | 0.0 | 0 | 0.0 |
| | | 1.2 | 282 | 2.1 | 0 | 0.0 |
| | | 1.5 | 163 | 0.4 | 80 | 1.2 |
| | | 2.0 | 34 | 0.1 | 148 | 0.9 |
| | 100 | 0.9 | 635 | 49.9 | 170 | 65.3 |
| | | 1.2 | 1459 | 13.4 | 336 | 29.6 |
| | | 1.5 | 929 | 2.6 | 483 | 8.7 |
| | | 2.0 | 415 | 0.8 | 284 | 1.7 |
| | 500 | 0.9 | 792 | 63.4 | 167 | 63.5 |
| | | 1.2 | 4344 | 65.3 | 597 | 66.9 |
| | | 1.5 | 7481 | 24.3 | 2270 | 66.1 |
| | | 2.0 | 4435 | 8.4 | 2115 | 13.6 |
| Netherlands | 10 | 0.9 | 298 | 14.4 | 0 | 0.0 |
| | | 1.2 | 243 | 1.8 | 0 | 0.0 |
| | | 1.5 | 183 | 0.5 | 133 | 2.0 |
| | | 2.0 | 0 | 0.0 | 22 | 0.1 |
| | 100 | 0.9 | 598 | 48.2 | 171 | 66.6 |
| | | 1.2 | 2333 | 20.1 | 362 | 30.7 |
| | | 1.5 | 953 | 2.7 | 419 | 7.5 |
| | | 2.0 | 854 | 1.5 | 0 | 0.0 |
| | 500 | 0.9 | 811 | 63.1 | 185 | 67.0 |
| | | 1.2 | 4857 | 68.5 | 604 | 67.3 |
| | | 1.5 | 8947 | 28.0 | 2123 | 65.1 |
| | | 2.0 | 3190 | 5.9 | 2960 | 18.1 |

No restrictions on access to symptomatic testing at clinics (i.e., all symptomatic individuals who sought testing at clinics would receive one if in stock) and high-risk household contacts of test-positive individuals are not tested. The average values tabulated are based on results from five independent simulations.

antivirals were distributed regardless of vaccination status. As such, increasing vaccination coverage did not lower antiviral need substantially (median 0.93-fold change (IQR = 0.70–1.00) when vaccination coverage increased from 10% to 90%). Conversely, the amount of antivirals distributed depends on $R_e$ (median 2.60-fold change (IQR = 0.97–4.35) when $R_e$ increases from 0.9 to 2.0), country demographics (median 1.72-fold change (IQR = 1.02–2.04) when distributing antivirals in Georgia relative to Zambia), testing rates (median 4.31-fold change (IQR = 1.49–5.77) when increasing from 100 to 500 tests/100 K/ day), and how tests were targeted (median 2.57-fold change (IQR = 1.52–4.55) when testing only high-risk as opposed to all symptomatic individuals).

## Impact of oral antivirals with over-the-counter self-tests

Unlike LMICs, over-the-counter Ag-RDTs were readily available in high-income countries during and after the emergency phase of the pandemic. In a separate analysis for the Netherlands, we assumed that over-the-counter Ag-RDTs for self-testing were widely available (i.e., with no-cap on availability) such that only 10% of symptomatic individuals seek clinic-provided testing directly. We also assumed that 80% of symptomatic individuals who did not seek clinic-provided testing may perform a self-test using over-the-counter Ag-RDTs instead. This effectively means that up to 82% of all symptomatic individuals would perform either a clinic-provided or over-the-counter self-test. All high-risk individuals who tested positive using self-tests would then seek reflexive testing at clinics on the same day to be administered antivirals. Clinic-provided testing would only be performed if they were still available under the average test availability of either 100 or 500 tests/100 K/day.

Under these assumptions, we found that in combination with clinic-provided testing rate of 500 tests/100 K/day, distribution of antivirals could avert 56–59% of severe cases and 67–70% of deaths on average, regardless of the epidemic intensity (Fig. 5). Reduction in infections due to antivirals was similarly modest and did not amount to more than an average of 13%. However, if mean clinic-provided testing rates fell to 100 tests/100 K/day, the mean proportion of severe cases and deaths averted would also drop precipitously to as low as 14% and 19%, respectively when $R_e \geq 1.5$. Across both testing rates and $R_e$, we found that one antiviral course was distributed for every 4–69 individuals for two 90-day epidemic waves in a year.

Since antivirals must be administered promptly upon a positive diagnosis, we also computed the proportion of high-risk, symptomatic individuals that would miss the treatment window if they had sought reflexive testing late. Regardless of clinical testing rate and $R_e$ for ≥90% of high-risk symptomatic individuals who were able to avert severe disease outcomes by antivirals to be treated with the drug, they must not seek reflexive testing at clinics (if reflexive testing is required) later than two days after being tested positive with over-the-counter self-tests (Figure S12).

## Effectiveness of test-and-treat strategies

To further compare the effectiveness of the test-and-treat strategies we investigated, we plotted efficiency curves of the number of severe cases averted by antivirals against the number of antivirals administered across all $R_e$ values and countries (Figure S13). As we assumed that there was no cap on antiviral availability, the limited test availability thus determines the number of antivirals distributed and in turn, the maximum number of severe cases averted by antivirals. We found that testing and treating test-positive, high-risk household contacts alongside the test-positive index individual ("Symptomatic + HR household" in Figure S13) was the least efficient test-and-treat strategy in our analyses. This is because allocating tests to screen high-risk household contacts, who may or may not be infected, under limited test availability reduced the number of tests that would otherwise have been used to identify symptomatic infected high-risk individuals

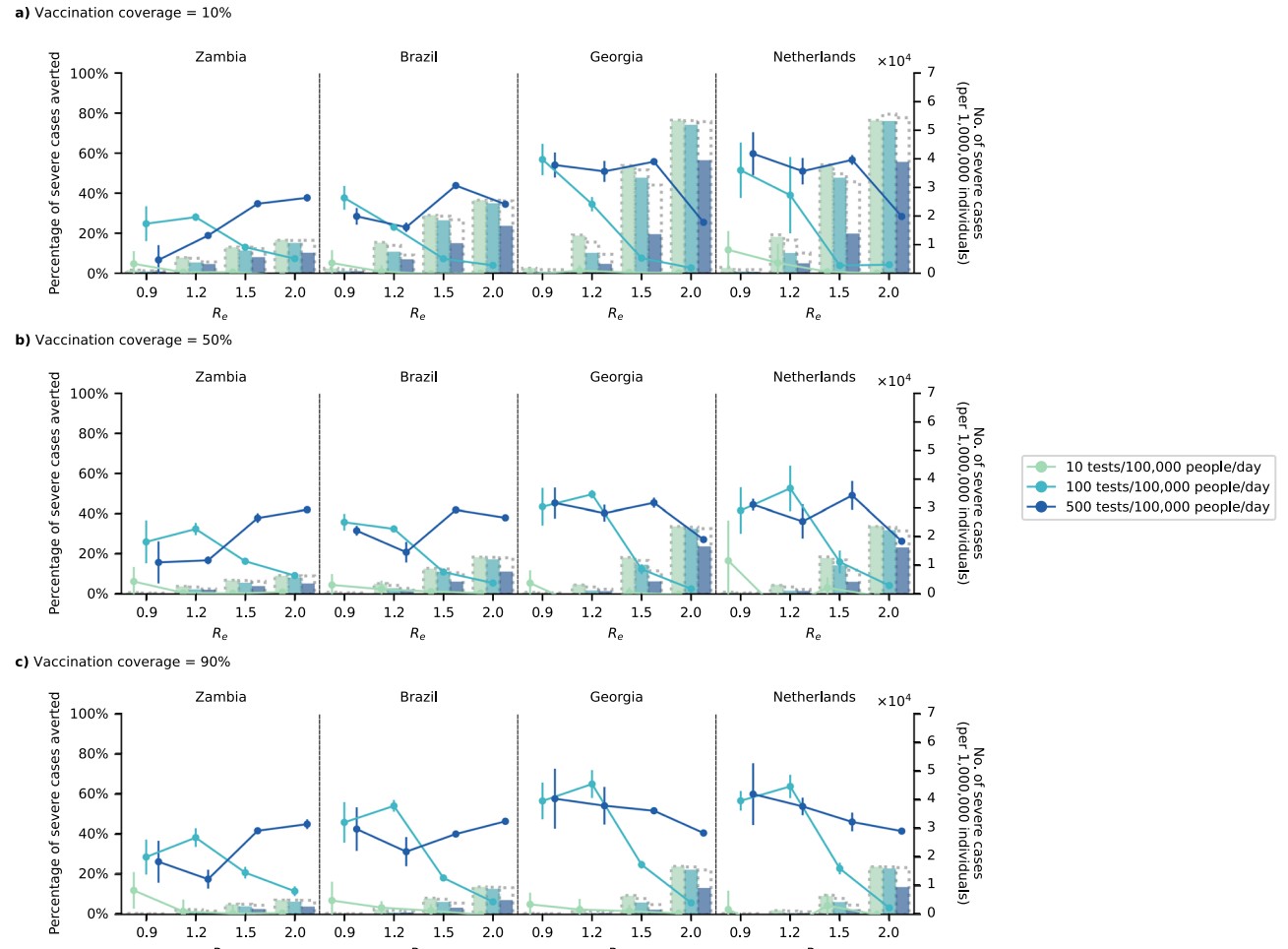

**Fig. 3 | Impact of test-and-treat on severe cases when restricting symptomatic testing to high-risk individuals only.** High-risk household contacts of test-positive individuals are not tested. All eligible high-risk individuals (i.e., ≥60 years of age or an adult ≥18 years with a relevant comorbidity) who tested positive were given a course of oral antivirals. Line plots (left y axis) show the mean percentage change (standard deviation denoted by error bars; n = 5 independent simulations) in severe cases relative to no distribution of antivirals under different levels of mean test availability (different shades of color) after a 90-day epidemic wave in a population of 1,000,000 individuals with **a** 10%, **b** 50%, and **c** 90% vaccination coverage for different epidemic intensities (measured by the initial effective reproduction number ($R_e$); x axis). Bar plots (right y axis) show the number of severe cases in each corresponding scenario. The dotted outline of each bar shows the number of severe cases of each scenario when no antivirals were distributed.

for antiviral administration. Restricting tests to high-risk individuals only ("HR symptomatic only") was similarly effective to no restriction in access to tests for all symptomatic individuals ("Symptomatic") as it is an essentially a workaround of the latter strategy to increase the number of high-risk infected individuals who are tested and treated under limited test-availability. In short, the greater the access high-risk individuals have to testing, the more likely they could be identified for timely treatment by antivirals. This could also be achieved when we test all symptomatic individuals but ensuring the wide availability of over-the-counter self-tests alongside large clinic-based test availability ("OTC self-test").

**Sensitivity analyses**

We performed several sensitivity analyses in Georgia, owing to the relatively greater impact of antivirals among the simulated LMICs, and investigated the extent to which our results may deviate under different key assumptions. First, unvaccinated individuals could have shared sociodemographic traits[32] and consequently vaccinated individuals would not necessarily be randomly distributed across the population. As an approximation, we assumed that vaccinated individuals, while still tiered by age, cluster among members from the same

household. Although reduction in infections remained similarly modest even when vaccinated individuals tended to be clustered (Figure S14), a larger proportion of severe cases were averted by antivirals (50% vaccinated: 8% (random) vs. 30% (clustered); 90% vaccinated: 14% (random) vs. 44% (clustered)) at the highest epidemic intensity simulated ($R_e$ = 2) but only if testing rates were large enough to support the distribution of antivirals (500 tests/100,000 people/day; Figure S15). However, the greater impact of antivirals on severe cases here is attributed to the increased number of severe cases stemming from vaccinated individuals being clustered (Figure S15B). We found that severe cases increased by 15–170% across all simulated scenarios if vaccinated individuals were clustered by households as opposed to being randomly assigned. This correspondingly led to greater oral antiviral demand as well with one antiviral course distributed for every 53–128 (5–104) persons per year if testing rate was 100 (500) tests/100 K/day. In short, while oral antivirals could alleviate the greater disease burden associated with clustering among vaccinated individuals, it is only facilitated by large enough testing rates and the need for greater antiviral supply. The more critical factor towards lowering severe cases is to minimize spatial bias among vaccinated individuals.

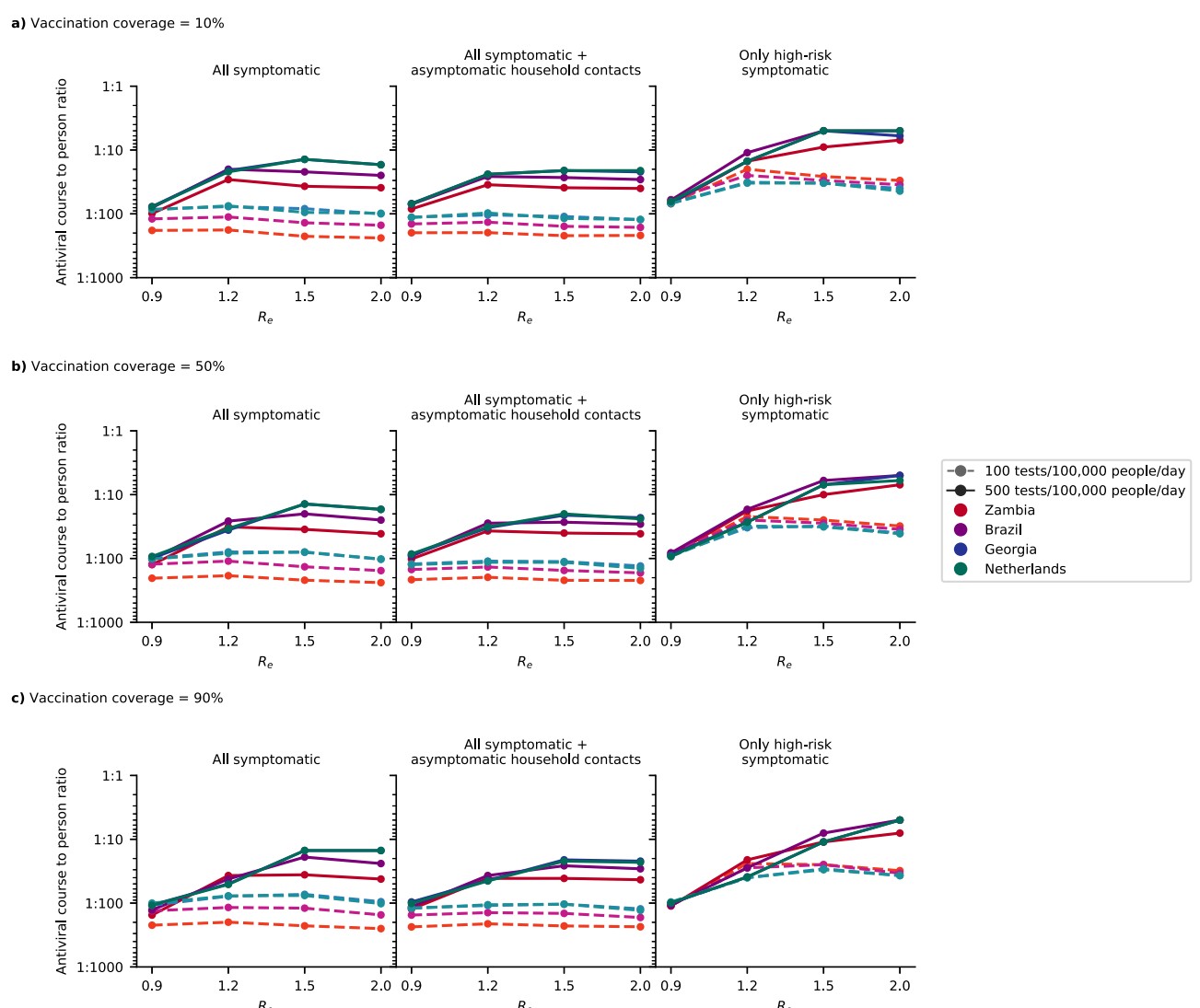

**Fig. 4 | Estimated need of oral antivirals.** Line plots show the ratio of estimated oral antiviral courses needed to number of people per year (expressed as 1 oral antiviral course per *n* number of individuals; assuming two epidemic waves a year) in simulated countries (color) under different simulated scenarios (i.e., testing rate at 100 or 500 tests/100,000 people/day (shading and linestyle) and distribution modality (left plot panel: test all symptomatic individuals who sought testing at clinics; middle plot panel: test all symptomatic individuals who sought testing as well as distributing clinic-provided self-tests to high-risk asymptomatic household contacts of test-positive individuals; right plot panel: test only high-risk symptomatic individuals who sought testing at clinics). All test-positive eligible high-risk individuals from clinic-provided testing would receive a course of oral antivirals. **a** 10%, **b** 50%, and **c** 90% vaccination coverage assumed for the simulated population.

Second, we had assumed low average estimates of vaccine effectiveness (i.e., 29% and 70% protection against infection and severe disease respectively). However, vaccine effectiveness can be improved by updating the vaccine strains to match circulating viruses or through booster shots. We repeated our simulations with vaccines conferring greater effectiveness, including known average protection against Delta-like (i.e., 52% and 96% protection against infection and severe disease respectively) and wild-type SARS-CoV-2 viruses (i.e., 75% and 97% protection against infection and severe disease)[19–21]. Similar to our original results for low vaccine effectiveness, use of antivirals could reduce transmissions in Georgia by up to ~20% but only if testing rates were high (500 tests/100 K/day; Figure S16). In contrast, the proportion of severe cases averted due to antivirals became increasingly uncertain (i.e., wider error bars in Figure S17). This was because improved vaccine effectiveness, on top of wider vaccination coverage, substantially reduced the number of severe cases. Nonetheless, regardless of vaccine effectiveness and coverage, meaningful reductions in severe cases by antivirals could only be achieved with higher

testing rates (≥100 tests/100 K/day) to support the administration of antivirals for infected high-risk individuals.

Third, we lowered the epidemic seeding condition from 1% to 0.1% such that antivirals were distributed and used by the population earlier akin to the situation where Paxlovid is readily available in certain countries. Although reduction in infections by antivirals continues to be achieved only at higher testing rates, if antivirals were distributed earlier (i.e., starting from a lower seeding condition), infections could be lowered by up to 30% even when test availability was 100 tests/100 K/day (e.g., At 100 tests/100 K/day and 50% vaccination coverage, only an average of 1% of infections were averted due to antivirals when $R_e$ = 1.5 if the seeding condition was set to 1% but increased to 21% if seeding proportion was lowered to 0.1%; Figure S18). The reduction in infections compounded the impact of antivirals on severe case reduction: the proportion of severe disease averted due to antivirals increased with improved outcomes at higher vaccination coverage (e.g., At 100 tests/100 K/day and 50% vaccination coverage, only an average of 4% of severe cases were averted due to antivirals when

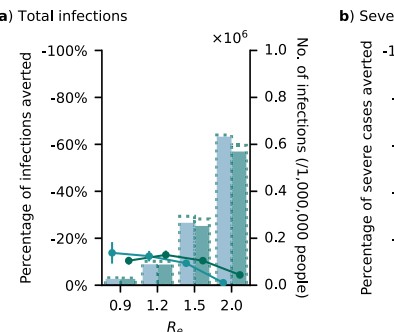
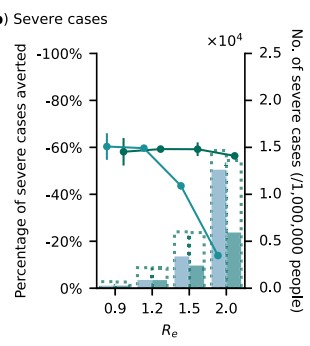
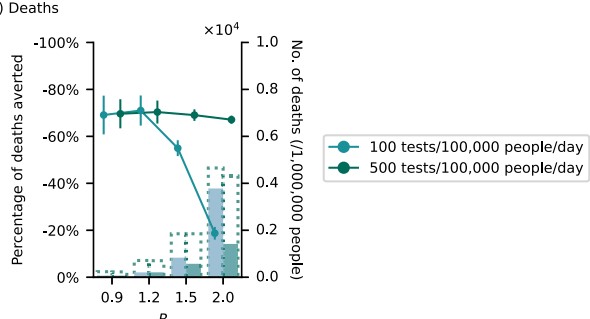

**Fig. 5 | Impact of test-and-treat in a high-income country (Netherlands) with wide availability of over-the-counter-self-tests.** No restrictions on access to symptomatic testing at clinics (i.e., all symptomatic individuals who sought testing at clinics would receive one if in stock) and high-risk household contacts of test-positive individuals are not tested. Over-the-counter antigen rapid diagnostic tests (Ag-RDTs) are assumed to be widely available with unlimited stocks. As such, we assumed that only 10% of symptomatic individuals would seek clinical testing directly while 80% of those who opted not to seek clinic-provided testing would perform self-testing using over-the-counter Ag-RDTs. All high-risk individuals who tested positive through self-testing would seek reflexive testing at clinics on the same day. All eligible high-risk individuals (i.e., ≥60 years of age or an adult ≥18 years with a relevant comorbidity) who tested positive at clinics, either directly or through reflexive testing, were given a course of oral antivirals. Line plots (left $y$ axis) show the mean percentage change (standard deviation denoted by error bars; $n = 5$ independent simulations) in **a** total infections, **b** severe cases and **c** deaths relative to no distribution of antivirals under different clinical testing rates (different shades of color) after a 90-day SARS-CoV-2 epidemic wave in a population of 1,000,000 individuals with 80% vaccination coverage for different epidemic intensities (measured by the initial effective reproduction number ($R_e$); $x$ axis). Bar plots (right $y$ axis) show the number of severe cases in each corresponding scenario. The dotted outline of each bar shows the number of severe cases of each scenario when no antivirals were distributed.

$R_e = 1.5$ if seeding condition was at 1% but increased to 47% if seeding proportion was lowered to 0.1%; Figure S19). The lower seeding condition also led to a fair proportion of severe cases averted at 10 tests/100 K/day but with large uncertainty (i.e., wider error bars in Figure S19A) and mostly only when $R_e < 1$ or at high vaccination coverage (90%). This suggests that the benefit of antivirals can be further augmented by early widespread adoption of test-and-treat programs.

Finally, the results above were predicated on crisis-period willingness-to-test behavior. However, in many countries, the willingness of people to test has waned substantially in the post-emergency phase of the pandemic. To investigate the consequences of this decline, we repeated our simulations for all countries assuming that the likelihood a symptomatic individual seek testing was 10% instead of the 65% assumed in the prior results (Figure S20). Low willingness to test would substantial reduce the impact of potential test-and-treat programs under all test availabilities, averting no >-10% of severe cases on average in any country for all simulated $R_e$ values and vaccination coverage.

## Discussion

Individual-level data on the effectiveness of antivirals for reducing severe disease[3,4] and the modeling work presented here highlight that substantial reductions in COVID-19 disease burden could arise from population-level test-and-treat programs. However, the low testing rates in the post-emergency phase of the pandemic represent a profound impediment for realizing the benefits of such programs. Most of the analyses described here focused on test availability as the functional constraint on the development of test-and-treat programs and this remains an issue in many LMICs. However, in many countries, regardless of socio-economic status, the willingness of people to pursue testing for respiratory virus disease is either low or declining and this presents challenges even when tests are available.

Given that antivirals are unlikely to have substantial impact on population-level transmission[5], if the main objective of testing is to maximize the distribution of antivirals to infected high-risk individuals, restricting clinic-based testing to only high-risk symptomatic individuals at testing rates of 100 tests/100 K/day could lead to 3.3–6.4-fold increase in proportions of severe cases averted relative to the default scenario where no restrictions to clinic-provided testing was imposed, provided that people are proactively seeking testing.

It is also possible to require asymptomatic, high-risk household contacts of test-positive symptomatic individuals to perform self-tests in order to initiate as many high-risk infected individuals to early antiviral treatment as possible. However, setting aside tests to screen high-risk household contacts under test availability constraints diminish the utility of tests that would have otherwise been used to test symptomatic individuals who sought testing. In turn, the proportion of severe cases and deaths averted due to antiviral distribution decrease by a relative factor of two to ten-fold under this strategy. A potential workaround could be to distribute antivirals to high-risk household contacts of test-positive individuals without the need to confirm if the high-risk contacts were infected themselves by testing. However, this would also increase the number of antiviral courses needed as well as result in wastage among individuals who were not infected. A cost effectiveness analysis could be performed to identify the most resource effective strategy but is beyond the scope of this work.

Assuming high willingness to test, ensuring the wide availability of over-the-counter self-tests could also lead to substantial reductions in severe cases (56–59%) and deaths (67–70%) at $R_e \geq 1.5$ (e.g., BA.1 or BA.5 variant-like events). However, if reflexive testing is needed for administration of antivirals, these reductions would only be possible if clinic-provided testing is maintained at the mean HIC rate of 500 tests/100 K/day. If clinical testing volumes drop to 100 tests/100 K/day, the expected reduction in severe cases and deaths attributable to antivirals would fall to only 14% and 19%, respectively in an epidemic wave initializing at $R_e = 2.0$ (e.g., BA.1 variant-like event).

Our results suggest that regardless of the (test and antiviral) distribution strategy, an effective test-and-treat program in any country requires large testing rates (>>100 tests/100 K/day) that are far beyond testing rates reported globally since 2023[9]. In turn, increasing vaccination is likely a more viable approach to lower severe cases than implementing large-scale test-and-treat programs. To compare the vaccination coverage and the resource requirements needed for test-and-treat to achieve the same reduction in disease burden, we computed the additional vaccination coverage needed to halve the number of severe cases at different $R_e$ under 10% starting vaccination coverage. We also estimated the equivalent number of tests and antivirals distributed to half the number of severe cases (Table 2). Across all countries and $R_e$, we estimated that an additional 24%-67% of the population must be vaccinated to reduce the number of severe cases

**Table 2 | Vaccination coverage and test-and-treat requirements to half the number of severe cases at 10% pre-existing vaccination coverage**

| Country | $R_e$ | Vaccination | Test-and-treat | |
|---|---|---|---|---|
| | | No. of vaccinated individuals per 1,000,000 people | No. of antivirals distributed per 1,000,000 people | Testing rate (tests/ 100,000 people/day) |
| Zambia | 1.2 | 496,192 | 18,436 | 545 |
| | 1.5 | 586,432 | 60,370 | 2185 |
| | 2.0 | 665,567 | 113,835 | 4383 |
| Brazil | 1.2 | 312,095 | 15,494 | 316 |
| | 1.5 | 437,067 | 82,093 | 1760 |
| | 2.0 | 566,068 | 197,465 | 4751 |
| Georgia | 1.2 | 253,988 | 9246 | 166 |
| | 1.5 | 317,855 | 52,499 | 721 |
| | 2.0 | 477,855 | 211,977 | 3471 |
| Netherlands | 1.2 | 240,271 | 8758 | 206 |
| | 1.5 | 319,695 | 49,952 | 701 |
| | 2.0 | 474,469 | 394,414 | 6802 |

For vaccination, the additional number of individuals per 1,000,000 people that must be vaccinated on top of the 10% pre-existing vaccination coverage is tabulated. For test-and-treat, there are no restrictions on access to symptomatic testing at clinics (i.e., all symptomatic individuals who sought testing at clinics would receive one if in stock) and high-risk household contacts of test-positive individuals are not tested.

by half without antivirals. Conversely, ~9000–400,000 courses of antivirals per 1,000,000 people would be needed to avert the same number of severe cases by antivirals for one epidemic wave. Furthermore, we estimated that ~200–7000 tests must be performed per 100,000 people per day to support the distribution of those antivirals. While these testing rates were achieved by some high-income countries during the COVID-19 pandemic, no countries are testing at anywhere near these rates in the post-emergency phase, suggesting that vaccination would likely be the more efficient option for reducing severe disease burden.

There have been other modeling efforts that estimated substantial reductions in disease burden by distributing antivirals to 20–50% of symptomatic infected individuals. However, from our analyses, doing so would also require testing rates that are far greater than 500 tests/100,000 people/day. First, Leung et al.[7] estimated that distributing antivirals to 50% of all symptomatic infected individuals regardless of risk status would only reduce hospitalizations by 10–13% in a population with high vaccination coverage (70–90%). For the Netherlands, we simulated a population with 80% vaccination coverage and large test availability, that included both clinic-based and over-the-counter self-tests, such that at least 50% of all symptomatic individuals were diagnosed. We estimated that 56–59% of severe cases could be averted if only high-risk symptomatic individuals were administered antivirals. When we reconfigured our simulations to now distribute antivirals to 50% of all symptomatic infected individuals, the proportion of severe cases averted lower to only 18% which is more in line with Leung et al.

Second, Matrajt et al. found that initiating 20% of infected individuals that were >65 years of age on antivirals daily could avert 32-43% of deaths in an Omicron-like wave ($R_e \geq 2$) for an unvaccinated population in LMICs such as Kenya and Mexico[5]. We had estimated that 31–62% of deaths could be averted at $R_e = 2$ at low (10%) vaccination coverage in LMICs but only if test availability was 500 tests/100 K/day and clinic-provided symptomatic testing were restricted to high-risk individuals, which would mean a daily average of 19–20% of high-risk infected individuals being initiated on treatment each day. If there are no restrictions on access to clinic-provided tests, testing rate must be

at least 750 tests/100 K/day to initiate 20% of infected >65 years on antivirals daily with >95% probability, indicating that the previous from Martrajt et al. predicated on very high testing rates.

Finally, Brault et al. estimated that 11% of hospitalizations could be averted if antivirals with 50% effectiveness were administered to half of all high-risk cases in Wallis and Futuna, where ~70% of individuals have at least two doses of vaccines, during an epidemic wave with a doubling time of 2–3 days[8]. In the closest scenario we had simulated (i.e., $R_e = 2$, 46% effectiveness of antivirals, 50% vaccination coverage and 500 tests/100 K/day), we estimated that severe cases could be reduced by 7% in Brazil (Fig. 2B), which has a similar demography to Wallis and Futuna (i.e., median age = 33 and 35 years in Brazil and Wallis and Futuna respectively; proportion of individuals ≥65 years = 10% and 13%, respectively). However, like the two preceding examples, this is only possible at testing rates that are many-fold higher than those performed in most LMICs both during and after the emergency phase of the pandemic.

There are limitations to our work: First, our simulations were based on the estimated effectiveness of nirmatrelvir–ritonavir. We did not consider the clinical benefits of other oral antivirals as nirmatrelvir–ritonavir was the most efficacious antiviral available during the development of this work.

Second, as a simplification, we assumed that individuals with pre-existing comorbidities that augment the risk of severe COVID-19 disease outcomes were randomly distributed across the population. The prevalence of certain comorbidities is known to correlate with socio-economic and demographic factors[33,34], resulting in the clustering of severe cases with similar socio-economic backgrounds. However, there is limited country-specific data on the prevalence and distribution of comorbidities across the population, especially for LMICs. We would also need to stratify the simulation population socio-economically which is beyond the scope of this study.

Third, we had assumed that clinical testing for disease and administration of treatment occur on the same day in our simulations. However, any practical barriers that limit timely access to antivirals (e.g., inadequate supply and distribution, limited access to healthcare providers, acceptance of antiviral therapy) can substantially reduce the estimated impact of test-and-treat programs[35]. As shown in Figure S12, even under a large test availability scenario (with self-tests), if administration of antivirals was delayed by >2 days, <20% of high-risk treated individuals received their antiviral courses within the 5 days post-symptom onset window when Paxlovid was reported to be efficacious. As such, even if testing rates could sufficiently support test-and-treat programs, delays in accessing antivirals, which had been reported in various LMICs[36], must be minimized for these programs to remain effective. Ideally, testing and treatment of infected patients should occur at the same clinical interaction.

Next, others have showed that with greater vaccine effectiveness against infection (60%), a high vaccination coverage (~70–80%) coupled with antivirals that have an effect in lowering transmissions could synergistically reduce infections in the population[5]. However, for only ~20% of infections to be averted in an Omicron-like wave (i.e., doubling time of 2–3 days[37]), the antiviral must block onward transmission completely after initiating treatment and 30% of symptomatic infected adults must be administered antivirals daily[5]. Even if an antiviral that is 100% effective in truncating transmissions exist and there was high willingness to test, the testing rate must at least be 764 tests/100 K/day to initiate 30% of symptomatic infected individuals to treatment daily with >95% probability based on our estimates.

Finally, we did not factor in changes to individual immunity levels due to previous infections or immune waning. As a simplification, we assumed that these effects have been implicitly captured by various initial $R_e$ values and were able to simulate epidemics with prevalence ranges similar to those reported during the spread of Omicron sub-variants BA.5 and XBB.1.5 (Figure S1). However, it is currently unclear

how changing immunity dynamics in the future could affect severe disease outcomes.

Taken all together, while test-and-treat programs have substantial theoretical utility for reducing population-level burden of disease, there remain fundamental challenges in terms of the availability of diagnostics and the willingness of people to seek testing in general and particularly within the relatively short window of effectiveness of the antivirals considered here. The potential benefits and resource requirements of test-and-treat programs must also be carefully considered if budget constraints make vaccination programs a competing interest.

## Methods

### The PATAT simulation model

PATAT creates an age-structured population of individuals within contact networks of multi-generational households, schools, workplaces, regular mass gatherings (e.g., religious gatherings) and random community settings with country-specific demographic data (see Supplementary Text). Epidemic simulations begin with 1% of the population infected with SARS-CoV-2 and compute transmissions between individuals across different contact networks each day. The computational flow of a PATAT simulation is summarized as follows: First, an age-structured population of agents is created. Close contact networks are subsequently created based on the given demographic data. The simulation is then initialized and iterates over a given period of time where each timestep corresponds to a day. The operations during each timestep encompass updating the disease progression of infected individuals, the status of isolated/quarantined agents, application of community testing strategies and computation of transmission events within contact networks.

PATAT implements a SEIRD epidemic model where the simulated population is distinguished between five compartments: susceptible, exposed (i.e., infected but is not infectious yet; latent phase), infected (which include the presymptomatic infectious period for symptomatic agents), recovered and dead. The infected compartments are further stratified by their presented symptoms, including asymptomatic, presymptomatic, symptomatic mild or severe. All symptomatic agents will also first undergo an infectious presymptomatic period after the exposed latent period. They will either develop mild symptoms who will always recover from the disease or experience severe infection which could either lead to death or recovery. As a simplification, PATAT assumes that all agents presenting severe symptoms are sufficiently isolated from the population (e.g., through hospitalization) that they are unlikely to contribute to further transmissions.

When an infectious agent $i$ comes into contact with a susceptible individual $j$, the probability of transmission ($p_{\text{transmission}, (i, j)}$) is given by:

$$p_{\text{transmission},(i,j)} = \beta \times \Phi_i \times f_c \times f_{\text{asymp},i} \times f_{\text{load},i} \times f_{\text{immunity},j} \times f_{\text{susceptiblity},j} \times \rho_i \times \rho_j$$

(1)

where $\beta$ is the base transmission probability per contact, $\Phi_i$ is the overdispersion factor modeling individual-level variation in secondary transmissions (i.e., superspreading events), $f_c$ is a relative weight adjusting $\beta$ for the network setting $c$ where the contact has occurred, $f_{\text{asymp},i}$ is the assumed relative transmissibility factor if infector $i$ is asymptomatic, $f_{\text{immunity},j}$ measures the immunity level of susceptible $j$ against the transmitted virus (i.e., $f_{\text{immunity},j} = 1$ if completely naive; $f_{\text{immunity},j} = 0$ if fully protected), $f_{\text{susceptiblity},j}$ is the age-dependent susceptibility of $j$, $\rho_i$ and $\rho_j$ are the contact rates of infector $i$ and susceptible $j$ respectively.

$\Phi_i$ is randomly drawn from a negative binomial distribution with mean of 1.0 and shape parameter of 0.45[38]. As evidence have been mixed as to whether asymptomatic agents are less transmissible, we conservatively assume there is no difference relative to symptomatic

patients (i.e., $f_{\text{asymp},i} = 1$). The age-structured relative susceptibility values $f_{\text{susceptiblity},j}$ are derived from odds ratios reported by Zhang et al.[22] (Table S1).

$\beta$ is determined by running initial test simulations with a range of values on a naïve population with no interventions that would satisfy the target reproduction number as computed from the resulting exponential growth rate and distribution of generation intervals[39]. $f_c$ is similarly calibrated during these test runs such that the transmission probabilities in households, workplaces, schools, and all other community contacts are constrained by a relative weighting of 10:2:2:1[23].

The total duration of infection since exposure depends on the symptoms presented by the patient and is comprised of different phases (i.e., latent, asymptomatic, presymptomatic, onset-to-recovery/death). The time period of each phase is drawn can be found in Table S1. For each infected individual, PATAT randomly draws a within-host viral load trajectory over the duration of infection, which impacts the sensitivity of Ag-RDTs[40], based on known distributions for Omicron BA.1[41]. Similar viral load trajectories were drawn for both asymptomatic and symptomatic infected individuals[42] using a stochastic model modified from the one previously developed by Quilty et al.[43] A baseline $Ct$ value ($Ct_{\text{baseline}}$) of 40 is established upon exposure. The infected agent becomes infectious upon the end of the latent period and their $Ct$ value is assumed to be ≤30. A peak $Ct$ value is then randomly drawn from a normal distribution (Table S1). Peak $Ct$ is assumed to occur upon symptom onset for symptomatic agents and one day after the latent period for asymptomatic individuals. Cessation of viral shedding (i.e., return to $Ct_{\text{baseline}}$) occurs upon recovery or death. PATAT assumes that the transition rate towards peak $Ct$ value should not be drastically different to that when returning to baseline upon cessation (i.e., there should be no sharp increase to baseline $Ct$ value after gradual decrease to peak $Ct$ value or vice versa). As such, the time periods of the different phases of infection are randomly drawn from the same quintile of their respective sample distribution. The viral load trajectory is then simulated by fitting a cubic Hermite spline to the generated exposed ($t_{\text{exposed}}$, $Ct_{\text{baseline}}$), latent ($t_{\text{latent}}$, $Ct_{\text{latent}} = 30$), peak ($t_{\text{peak}}$, $Ct_{\text{peak}}$) and cessation values ($t_{\text{recovered/death}}$, $Ct_{\text{baseline}}$). The slope of the fitted curve is assumed to be zero for all of them except during $t_{\text{latent}}$ where its slope is assumed to be $\frac{Ct_{peak} - Ct_{baseline}}{t_{peak} - t_{exposed}}$. PATAT then uses the fitted trajectory to linearly interpolate the viral load transmissibility factor $f_{\text{load},i}$)) of an infectious agent $i$ assuming that they are twice as transmissible at peak $Ct$ value (i.e., $f_{\text{load}} = 2$) relative to when they first become infectious (i.e., $Ct$ value = 30; $f_{\text{load}} = 1$).

Unlike PCR which is highly sensitive due to prior amplification of viral genetic materials, the sensitivity of Ag-RDT will depend on the viral load of the tested patient. While the specificity of Ag-RDT is assumed to be 98.9%, its sensitivity depends on the $Ct$ values of the tested infected agent: $Ct > 35$ (0%); 35−30 (20.9%); 29−25 (50.7%); $Ct \leq 24$ (95.8%)[40].

We assumed that agents would change their behavior when (i) they start to present symptoms and go into self-isolation (10% compliance assumed, 71% endpoint adherence)[27]; (ii) they test positive and are isolated for 10 days (50% compliance assumed, 86% endpoint adherence)[27]; or (iii) they are household members (without symptoms) of positively-tested agents and are required to be in quarantine for 14 days (50% compliance assumed, 28% endpoint adherence)[27]. Once an agent goes into isolation/quarantine, we linearly interpolate their probability of adherence to stay in isolation/quarantine over the respective period. Given the lack of infrastructure and resources to set up dedicated isolation/quarantine facilities in many LMICs, we assumed that all isolated and quarantined individuals would do so at home.

We simulated 90-day epidemic waves in a community of 1,000,000 individuals using demographic data collected from three LMICs (i.e., Brazil, Georgia, Zambia) and the Netherlands as a HIC counterpart. We simulated different vaccination coverage (10%, 50%,

and 90%) for all countries for comparability. In the separate analysis examining how widespread availability of over-the-counter self-tests could impact test-and-treat programs in HICs, we assumed that 80% of the population was vaccinated in the Netherlands based on estimates on July 2022[44], which is largely comparable to other HICs[45]. As a simplification, we assumed that vaccination protection rates against infection and severe disease were 29% and 70%, respectively, which were based on the more conservative, lower average estimates of vaccine effectiveness against BA.1 across different vaccines (i.e., mRNA and ChAdOx1 nCoV-19 vaccine) and doses (i.e., 1–3 doses)[19–21]. We did not assume a specific protection rate against death since the referenced studies had reported effectiveness estimates against severe disease outcomes which include hospitalization and/or death. Nonetheless, protection of deaths is implicitly accounted for since individuals could only die from COVID-19 if they had progressed to severe disease in the model.

### Diagnostic testing

In the model, individuals with symptomatic COVID-19 have a probability of seeking testing at a healthcare facility. We also estimated symptomatic testing demand from individuals without COVID-19 who sought clinic-provided testing (e.g., individuals who present with similar respiratory symptoms): Based on the range of test positivity rates reported by various countries during the second-half of 2021 (when community testing was assumed to still be prevalent in most countries)[45], we assumed that test positivity rate was 10% at the start as well as end of an epidemic wave, and a 20% test positivity rate at the peak, linearly interpolating the demand for periods between these time points[11,12].

We also simulated scenarios where household contacts of clinic-provided positively-tested individuals were given Ag-RDTs for self-testing for three consecutive days following the positive clinical test of the latter. Adherence (likelihood) to testing by asymptomatic household contacts was assumed to decrease linearly to 50% by the third day. We also simulated an alternative test distribution strategy where we restricted clinic-provided symptomatic testing to high-risk individuals only.

We performed simulations under three levels of average test availability at healthcare clinics: 10 (mean LMIC testing rate as of Q2/2022)[9], 100 and 500 (mean HIC testing rate as of Q2/2022)[9] tests/100 K/day. Regardless if symptomatic individuals choose to self-isolate, after $\tau_{\text{delay, symp-test}}$) days from symptom onset, the symptomatic agent may decide to get tested with a Bernoulli probability of $p_{\text{symp-test}}$). PATAT assumes that agents who have decided against symptomatic testing (i.e., failed Bernoulli trial) or received negative test results will not seek symptomatic testing again. We assumed that average $p_{\text{symp-test}} = 65\%$ on average based on surveys of test-seeking behavior during the COVID-19 pandemic[30,31]. In other words, there is an average 65% chance that an individual with mild symptom would seek clinic-provided testing and were only tested if there were available test stocks. We lowered $p_{\text{symp-test}}$ to 10% in a sensitivity analysis to estimate the impact of test-and-treat programs under waning willingness to test.

In the separate analysis where over-the-counter self-tests were available in the Netherlands, we assumed that only 10% of mild symptomatic individuals in the Netherlands would seek clinic-provided testing upon symptom onset based on average daily testing rates reported by all Dutch municipal health services in 2021-Q1/2022 (i.e., approximately up to the end of the Omicron BA.1 wave; 7551 tests/100 K/day) and Q2/2022 (post Omicron BA.1 wave; 641 tests/100 K/day)[46]. We assumed that 50% or 80% of individuals who opted not to seek clinic-provided testing would perform a self-test using an over-the-counter Ag-RDT. We assumed that all high-risk individuals who tested positive would then seek reflexive testing at clinics to be disbursed an antiviral course.

### Oral antivirals

Regardless of their vaccination status (per WHO guidance)[47], all high-risk individuals who tested positive within five days after symptom onset were eligible for a course of antiviral therapy[3,4]. We did not impose any caps on antiviral availability as we wanted to estimate the potential number of antiviral courses needed and thus their maximum achievable impact on epidemic outcomes under different levels of test availability and antiviral distribution strategies. We did not factor any risk reduction in transmissions or deaths given the lack and low certainty of evidence of the impact of oral antivirals on protection against transmission and mortality respectively[47]. However, individuals could only die from COVID-19 if they had progressed to severe disease in our model.

For individuals who were treated with antivirals that were simulated to result in severe disease, we performed a Bernoulli trial with the probability of averting severe disease (i.e., 46%), provided that they were currently in the presymptomatic phase or were experiencing mild disease. If the Bernoulli trial succeeded, we re-simulated their disease progression and within-host viral dynamics using the procedures above but now under the assumption that they would develop only mild disease and conditioning that the maximum viral load is lower than before. Changes were only made to the upcoming phases of disease progression from the then current phase of infection. The average recovery period of having mild symptoms was assumed to be 5.4 days as opposed to 18.1 days when presenting severe disease[41,48]. In turn, while we did not parameterize the impact on transmission reduction by antivirals, the shortened recovery, and thus infectious period as well as lower maximum viral load of individuals who were effectively treated could result indirect reduction in onward transmissions.

We performed five independent simulations for each combination of parameters described above. All key parameters are tabulated in Table S1. Further details of PATAT are described in Han et al.[11,12] and the Supplementary Information. The PATAT model source code is available at https://github.com/AMC-LAEB/PATAT-sim.

### Reporting summary

Further information on research design is available in the Nature Portfolio Reporting Summary linked to this article.

## Data availability

Data on global testing rates were downloaded from https://www.finddx.org/covid-19/test-tracker. All data relevant to the study are included in the Article, the Supplementary Information and the GitHub repository (https://github.com/AMC-LAEB/PATAT-sim).

## Code availability

The PATAT model source code and custom code used to analyzed our simulation data are available at https://github.com/AMC-LAEB/PATAT-sim and https://github.com/AMC-LAEB/PATAT-sim/blob/main/projects/av_therapeutics/han-et-al_av_therapeutics.ipynb respectively.

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

## Acknowledgements

This work was supported by the European Research Council [NaviFlu 818353 to A.X.H. and C.A.R.], the National Institutes of Health [5R01AI132362-04 to C.A.R.] and the Dutch Research Council (Nederlandse Organisatie voor Wetenschappelijk Onderzoek) [Vici 09150182010027 to C.A.R.]. This work was supported by the Rockefeller

Foundation, and the Governments of Germany, Canada, UK, Australia, Norway, Saudi Arabia, Kuwait, Netherlands and Portugal [all authors]. The authors are pleased to acknowledge that all computational work reported in this paper was performed on the Shared Computing Cluster which is administered by Boston University's Research Computing Services (www.bu.edu/tech/support/research/).

## Author contributions

Conceptualization, validation, writing—review & editing: A.X.H., E.H., S.C., B.R., B.E.N., C.A.R. Methodology, investigation, data curation, writing—original draft, visualization: A.X.H., B.E.N., C.A.R. Software, formal analysis: A.X.H. Resources, supervision, project administration: B.E.N., C.A.R. Funding acquisition: E.H., S.C., B.R., B.E.N., C.A.R.

## Competing interests

E.H., S.C., B.R., and B.E.N. declare that they are employed by FIND, the global alliance for diagnostics. The remaining authors declare no competing interests.
