## [Peer Review File · Nature Communications]

REVIEWER COMMENTS

Reviewer #1 (Remarks to the Author):

This paper ascertains the impact of test-and-treat strategies on the epidemiology of SARS-CoV-2 and the occurrence of severe cases in a LMICs setting. Using an agent-based model, the authors show that test-and-treat strategies would only have a limited impact in LMICs due to the low testing rates. Finally, the authors evaluate the impact of changing the target group for treatment as well as how the interplay between vaccination and treatment can shape the impact of test-and-treat interventions. Despite the wider availability of COVID-19 treatments, few studies have evaluated the impact of test-and-treat strategies, especially in LMIC settings where vaccination coverages are lower. The authors hence tackle a relevant question of public health importance. The methodology is appropriate to tackle such questions and the conclusions are well supported by the analyses. Finally, the manuscript is very clear and pleasant to read.

Major comments:

Though the questions tackled by the authors are very sound and relevant, I feel the current conclusions of the manuscript are quite scarce in terms of actual impact and public health implications. The main conclusion I get from the abstract is that for test-and-treat strategies to work, one needs to test, which is always good to show but feels a bit expected. I included a couple remarks below that might be useful to the authors to strengthen the current work.

1. Benefits of test-and-treat impacted by vaccine coverage.

One of the conclusions of the manuscript is that increasing vaccination coverage increases the benefits of test-and-treat (line 165). This was quite counter-intuitive for me. I wondered whether this was due to the fact that the authors are reported results in terms of in terms of proportion of severe cases averted. How does this trend hold when looking at the absolute number of severe cases averted? Is this just a result from the fact that higher vaccination coverage will decrease the effective reproduction number and hence result in less overall infections / severe cases? I imagine the authors might already have everything from their simulations to look at the absolute reduction in severe cases. This would be an easy addition to better interpret and understand these trends.

2. Effectiveness of test-and-treat strategies.

It is interesting to compare how different strategies translate in terms of antiviral needs (Figure 5) given that resources are likely to be an important element when comparing strategies. The way results are currently presented however does not enable to conclude which one is the most effective. One strategy may require less antivirals, but it may also result in a lower impact, which the figure currently does not depict. To do so, the authors could for example (i) look at the ratio between the number of severe cases averted per inhabitant and the antiviral course to person ratio or (ii) depict the number of severe cases averted as a function of the antiviral resources required under a given strategy. This could be seen as an equivalent of efficiency curves / efficiency frontiers that are traditionally used in health economics analyses and would better enable to compare the impact of these different strategies.

3. Relative benefits of test-and-treat vs vaccination.

In LMICs, the economic cost associated with these different public health interventions is likely to be an important element for their implementation. The authors find a limited impact of test-and-treat strategies under the current testing capacities in LMICs. Given the expected impact of vaccination on COVID-19 burden and the now wider availability of vaccine doses, I was left wondering which of vaccination vs test-and-treat (targeted towards at risk individuals) would be a better way to allocate resources. Given that the price associated with vaccination and test-and-treat are different, it could be valuable to understand / discuss which strategy would be the most cost effective to maximize severe cases reduction.

Other comments:

1. Many references haven't been properly compiled (line 69, 149, 156, 163, 217-218, 219, 248, 264, 282, 479).

2. The figures are overall of high quality. I have a couple suggestions that the authors may consider to improve readability and interpretability.

- In most figures, countries are ordered alphabetically. I would consider ordering them by increasing income level (i.e. Zambia-Brazil-Georgia-Netherlands) to better depict how the results are impacted by increasing income level.

- For figure 1b, there is no color scale for the heatmaps.

- It did not strike me at first that most figures were depicting the percentage of severe cases / infections averted due to test-and-treat strategies. To increase understanding by a less quantitative audience, I would consider renaming the y-axis by "Percentage of severe cases averted" instead of "Percentage Δ in severe cases".

- The color scheme in figures 2-3-4 is pretty. However, I found it a bit hard to understand to trend with increasing testing capacities with the current color schemes that better emphasizes the difference between countries that between testing intensities. I'm wondering whether trying to simply use 3 colours (for each of the different testing intensities) would facilitate interpretation.

- In most figures, the linetype (e.g. dashed, plain, pointed) used to depict different testing rates is hard to get from the legend.

3. Line 142: Should "Figure 2B" rather be "Figure S3B"?

4. I was wondering how to exactly interpret the reproduction number introduced by the authors. The authors mention it is a combination of population immunity, intrinsic transmissibility, and public health interventions. From the methods, it seems that the transmission rate beta is estimated for a given value of the reproduction number to match the expected growth rate in the absence of interventions. Does this mean that this is done in the absence of vaccination in the population? Or is this procedure repeated for any vaccine coverage considered in the scenario? If not, this might be just fine, but I would clarify that this is the expected reproduction number without vaccination and testing when introducing it. This would also make it clearer that these different scenarios are actually associated with different

effective reproduction numbers at the beginning of the epidemic (because of different vaccine coverages and testing rates).

5. Figure S1 does not show that we are more likely to detect a case when the age distribution is older given that the only thing depicted is the probability to detect symptomatic individuals (line 124-125). One way to do this could be to generate the same graphs but looking at the overall probability for an infected individual to be detected (and not conditional on being symptomatic).

6. The contextualization of the results obtained in LIMCs with regards to the impact of similar campaigns in HICs is valuable. The intervention scenario used for the Netherlands (in terms of vaccine coverage and testing) is however different from the one used for the LMICs. I understand the motivation of using a realistic scenario for the Netherlands, but I find it harder to compare the results and understand whether the observed differences are attributable to the different demographic structures or interventions considered. I am wondering whether also presenting results for Netherlands with the same intervention as for the LMICs (10-90% vaccine coverage, and a 10 to 500 testing rate) could clarify this.

7. If I understood correctly, treatments are distributed to individuals regardless of their vaccination status. Given the strong protection already conferred by vaccination against severe disease, could restricting the distribution of treatment to high-risk unvaccinated individuals largely increase the impact of the intervention?

8. At the beginning of the epidemic wave when infection rates are lower, I expect that testing will not saturate under any testing rate regime explored. However, as the epidemic progresses, I would expect that testing and treatment are “saturating” (i.e. the number of tests performed increases at the beginning of the epidemic and then plateaus when reaching the maximum capacities). When reaching this saturation (or maybe even at the beginning of the outbreak when expecting a larger wave), changing the definition of high-risk individuals (by restricting this group) might increase the overall impact. This could be discussed. The definition of high-risk individuals is indeed currently quite broad and the risk of developing a severe form of the disease upon infection is for example way higher in 80 y.o. than in 60 y.o.

Reviewer #2 (Remarks to the Author):

This modeling study evaluated the effectiveness of COVID-19 test and treat (via antivirals) strategies on reducing severe infections in low- and middle-income countries (LMIC). Using an agent-based model (ABM) framework, the authors demonstrate that significant reductions in severe cases with antivirals are only possible by substantially increasing testing rates in LMIC and/or restricting testing and availability of antivirals to high-risk populations only. This is a well-written and informative paper that may be useful for direction allocation of limited COVID-19 testing and treatment resources in LMIC for current and future health emergencies. However, there have been other notable studies in this field. This current manuscript would benefit from an overview of previous studies and what this paper adds to the existing literature. Additional comments are provided below.

Comments:

1. Introduction: Please describe other notable studies in this area (e.g., references [1-3])¹⁻³ and discuss what this current study adds to the literature.

2. Methods: More detail needed on modeling the impact of antivirals on transmission reduction

3. Results (lines 83-89) and Discussion (lines 460-464): What are the potential consequences of consolidating the effects of immunity from previous infection into instantaneous reproduction number (R_t)? Is this expected to yield bias or reduce precision?

4. Results: The finding that increasing vaccination coverage does not change the impact of antiviral distribution on infections averted appears counterintuitive given that vaccines are highly protective against severe infection. Can the authors provide some rationale for this finding?

5. Results. While 2-dose mRNA vaccines are less protective against general infection, studies have found that booster doses are protective against general infection from omicron.⁴ Were booster doses considered in this study? The authors may consider tying this into the sensitivity analyses results on lines 310-323.

6. Results (lines 265-266). Several studies have demonstrated that vaccines are protective against severe SARS-CoV-2 infection.⁵ If antivirals were to be prioritized, should vaccination status not be accounted for?

7. Discussion. In this section, it would be helpful to link different values of the instantaneous reproduction number (R_t) to known events. For example, on line 361-362 the authors could phrase this as "..., could average severe cases by up to 65% in LMICs experiencing an epidemic wave that initialized at $R_t \leq 1.2$ (e.g., BA.5 variant-like event).

8. Discussion (lines 377-380). Another potential strategy is simply testing all high-risk individuals in a household if any individual within that household is infected. Previous research has showed such strategies have a substantially higher probability of detecting infected cases.⁶

9. Discussion (lines 433-436). I suggest the authors perform a sensitivity analysis to the assumption that limits timely access to antivirals, as this is certainly a barrier in many LMIC.

10. Introduction/Discussion. There are currently strategies in place for prioritizing administration of Covid-19 treatment (e.g., Paxlovid; <https://www.cdc.gov/coronavirus/2019-ncov/hcp/clinical-care/outpatient-treatment-overview.html>). However, strategies that have not been implemented are not discussed in the context of this study.

11. Methods (diagnostic testing, lines 589-590). Please provide more detail on how this was estimated, including the role of access ability to healthcare facilities.

12. References: There are several instances of references not working/linking.

13. Appendix (household, lines 33-34). Please provide more explanation and detail on how households were ordered to implicitly approximate neighborhood proximity.

14. Appendix (model validation, lines 80-92). Why was the ABM only validated in one country prior to March of 2021? Given there have been several high-profile variants in the last 2 years (including Delta and Omicron variants), the authors should consider validating this model in more relevant settings.

15. Appendix (Table S1). In addition to providing distributional assumptions for input parameters, the authors may consider providing the median (IQR) or mean (SD) of these distributions. This will be more understandable to lay audiences and makes it easier to evaluate underlying assumptions.

References

1. L M, Er B, Ms C, D D, H J. Could widespread use of antiviral treatment curb the COVID-19 pandemic? A modeling study. *BMC infectious diseases*. 2022;22(1). doi:10.1186/s12879-022-07639-1
2. Leung K, Jit M, Leung GM, Wu JT. The allocation of COVID-19 vaccines and antivirals against emerging SARS-CoV-2 variants of concern in East Asia and Pacific region: A modelling study. *Lancet Reg Health West Pac*. 2022;21:100389. doi:10.1016/j.lanwpc.2022.100389
3. Brault A, Tran-Kiem C, Couteaux C, et al. Modelling the end of a Zero-COVID strategy using nirmatrelvir/ritonavir, vaccination and NPIs in Wallis and Futuna. *Lancet Reg Health West Pac*. 2023;30:100634. doi:10.1016/j.lanwpc.2022.100634
4. Rennert L, Ma Z, McMahan C, Dean D. Covid-19 vaccine effectiveness against general SARS-CoV-2 infection from the omicron variant: A retrospective cohort study. *PLoS Global Public Health*. 2023;3(1):e0001111. doi:<https://doi.org/10.1371/journal.pgph.0001111>

5. Bloomfield LE, Ngeh S, Cadby G, Hutcheon K, Effler PV. SARS-CoV-2 Vaccine Effectiveness against Omicron Variant in Infection-Naive Population, Australia, 2022 - Volume 29, Number 6—June 2023 - Emerging Infectious Diseases journal - CDC. doi:10.3201/eid2906.230130

6. Rennert L, McMahan C, Kalbaugh CA, et al. Surveillance-based informative testing for detection and containment of SARS-CoV-2 outbreaks on a public university campus: an observational and modelling study. *The Lancet Child & Adolescent Health*. 2021;5(6):428-436. doi:10.1016/S2352-4642(21)00060-2

Reviewer #3 (Remarks to the Author):

This is a very well-presented and comprehensive study detailing the assessment of the impact of a test-to-treat program with rapid tests and antivirals on SARS-CoV-2 infection outcomes. The analyses are largely appropriate, with considerable attention being paid to determining the sensitivity of the results to variations in key parameters.

However, the analysis focuses primarily on scenarios in which a relatively small proportion of the population initially infected (1% and lower in sensitivity analyses) with a 90-day epidemic wave subsequently following, as may have occurred during the pandemic phase; population immunity (beyond age-specific susceptibility and vaccine-derived immunity) is not accounted for in the model due to complexities in individual infection and vaccination histories between countries and over time. Given it appears that we are in the post-crisis, “endemic” phase, it would be prudent to assess test-to-treat in additional scenarios with a higher prevalence of infection, varying between 1-10%, but with R averaging 1 over a longer period (i.e., as observed by the UK ONS Covid Infection Survey in 2022-2023 with the Omicron subvariant waves

<https://www.ons.gov.uk/peoplepopulationandcommunity/healthandsocialcare/conditionsanddiseases/bulletins/coronaviruscovid19infectionsurveypilot/24march2023>), and low severity of infection (due to population immunity). It is not clear how future SARS-CoV-2 dynamics will proceed given the waning of infection and vaccine-derived immunity as well as the emergence of variants, though a high-prevalence endemic state with lower severity is possible and hence test-to-treat should be assessed in this context. This should also be talked about in the discussion.

Methods and supplement: Unless I have missed it, it is not stated how vaccines protect against death (infection and severe disease are noted on line 93; it would be good to have these in the table). This should be made explicit as it is a key parameter in the model.

Additional comments:

There are several missing references in the text (Error! Reference source not found.)

Line 486: dates should be referenced clearly (what does “last two years” of the pandemic refer to?) throughout the text.

Reviewers' comments in blue; authors' response in black.

Reviewer #1 (Remarks to the Author):

This paper ascertains the impact of test-and-treat strategies on the epidemiology of SARS-CoV-2 and the occurrence of severe cases in a LMICs setting. Using an agent-based model, the authors show that test-and-treat strategies would only have a limited impact in LMICs due to the low testing rates. Finally, the authors evaluate the impact of changing the target group for treatment as well as how the interplay between vaccination and treatment can shape the impact of test-and-treat interventions. Despite the wider availability of COVID-19 treatments, few studies have evaluated the impact of test-and-treat strategies, especially in LMIC settings where vaccination coverages are lower. The authors hence tackle a relevant question of public health importance. The methodology is appropriate to tackle such questions and the conclusions are well supported by the analyses. Finally, the manuscript is very clear and pleasant to read.

We thank the reviewer for their constructive comments.

Major comments:

1. Though the questions tackled by the authors are very sound and relevant, I feel the current conclusions of the manuscript are quite scarce in terms of actual impact and public health implications. The main conclusion I get from the abstract is that for test-and-treat strategies to work, one needs to test, which is always good to show but feels a bit expected. I included a couple remarks below that might be useful to the authors to strengthen the current work.

Beyond the fact that “we need to test for test-and-treat strategies to work”, we have now revised our manuscript to strengthen the key conclusions summarized in the abstract.

Line 26: “We find that in the post-emergency phase of the pandemic, in countries where low testing rates are driven by limited testing capacity, significant population-level impact of test-and-treat programs can only be achieved by both increasing testing rates and prioritizing individuals with greater risk of severe disease. However, for all countries, significant reductions in severe cases with antivirals are only possible if testing rates were substantially increased with high willingness of people to seek testing. Comparing the potential population-level reductions in severe disease outcomes of test-to-treat programs and vaccination shows that test-and-treat strategies are likely substantially more resource intensive requiring very high levels of testing (>>100 tests/100,000 people/day) and antiviral use suggesting that vaccination should be a higher priority”.

2. Benefits of test-and-treat impacted by vaccine coverage. One of the conclusions of the manuscript is that increasing vaccination coverage increases the benefits of test-and-treat (line 165). This was quite counter-intuitive for me. I wondered whether this was due to the fact that the authors are reported results in terms of in terms of proportion of severe cases averted. How does this trend hold when looking at the absolute number of severe cases averted? Is this just a result from the fact that higher vaccination coverage will decrease the effective reproduction number and hence result in less overall infections / severe cases? I imagine the authors might already have everything from their simulations to look at the absolute reduction in severe cases. This would be an easy addition to better interpret and understand these trends.

This is a good point. We have now clarified that the effective reproduction number should be interpreted as *“as the collective outcome of population immunity from previous infections, intrinsic transmissibility of the variant virus as well as effects of any existing any public health interventions other than vaccination and oral antivirals”* (line 92).

We reported the proportion of severe cases averted by antiviral use relative to no distribution of antivirals under the same vaccine coverage as the measure of antiviral benefit. As the reviewer correctly pointed out, wider vaccination coverage lowers the effective reproduction number and in turn, leads to lower number of infections and severe cases averted by distribution of antivirals. We have now included the absolute number of severe cases averted in Table 1 as well as revised lines 177-185:

“At testing rates of ≤ 10 tests/100,000 people/day, use of antivirals made negligible contributions to reducing severe disease at all levels of vaccine coverage. At testing rates ≥ 100 tests/100,000/people/day, higher vaccination coverage was associated with a smaller absolute number of severe cases averted by antivirals. However, at higher testing rates, the proportion of severe cases averted by antivirals relative to no distribution of antivirals is larger at higher vaccination coverage. This is because as infections decrease with higher vaccination coverage, a greater percentage of severe cases could also be detected and treated by antivirals assuming that the quantity of test availability is a constraining factor and that demand in low vaccination scenarios would exceed supply.”

3. Effectiveness of test-and-treat strategies. It is interesting to compare how different strategies translate in terms of antiviral needs (Figure 5) given that resources are likely to be an important element when comparing strategies. The way results are currently presented however does not enable to conclude which one is the most effective. One strategy may require less antivirals, but it may also result in a lower impact, which the figure currently does not depict. To do so, the authors could for example (i) look at the

ratio between the number of severe cases averted per inhabitant and the antiviral course to person ratio or (ii) depict the number of severe cases averted as a function of the antiviral resources required under a given strategy. This could be seen as an equivalent of efficiency curves / efficiency frontiers that are traditionally used in health economics analyses and would better enable to compare the impact of these different strategies.

We have now included the efficiency curves in Figure S13 and discussed how different strategies compared against each other based on our analyses:

Line 284: *“To further compare the effectiveness of the test-and-treat strategies we investigated, we plotted efficiency curves of the number of severe cases averted by antivirals against the number of antivirals administered across all R_e values and countries (Figure S13). As we assumed that there was no cap on antiviral availability, the limited test availability thus determines the number of antivirals distributed and in turn, the maximum number of severe cases averted by antivirals. We found that testing and treating test-positive, high-risk household contacts alongside the test-positive index individual (“Symptomatic + HR household” in Figure S13) was the least efficient test-and-treat strategy in our analyses. This is because allocating tests to screen high-risk household contacts, who may or may not be infected, under limited test availability reduced the number of tests that would otherwise have been used to identify symptomatic infected high-risk individuals for antiviral administration. Restricting tests to high-risk individuals only (“HR symptomatic only”) was similarly effective to no restriction in access to tests for all symptomatic individuals (“Symptomatic”) as it is an essentially a workaround of the latter strategy to increase the number of high-risk infected individuals who are tested and treated under limited test-availability. In short, the greater the access high-risk individuals have to testing, the more likely they could be identified for timely treatment by antivirals. This could also be achieved when we test all symptomatic individuals but ensuring the wide availability of over-the-counter self-tests alongside large clinic-based test availability (“OTC self-test”).”*

4. Relative benefits of test-and-treat vs vaccination. In LMICs, the economic cost associated with these different public health interventions is likely to be an important element for their implementation. The authors find a limited impact of test-and-treat strategies under the current testing capacities in LMICs. Given the expected impact of vaccination on COVID-19 burden and the now wider availability of vaccine doses, I was left wondering which of vaccination vs test-and-treat (targeted towards at risk individuals) would be a better way to allocate resources. Given that the price associated with vaccination and test-and-treat are different, it could be valuable to understand / discuss which strategy would be the most cost effective to maximize severe cases reduction.

Cost effectiveness is outside the scope of this study which primarily examines how testing rates and strategies impact the benefits of prospective test-and-treat programs under various levels of vaccination coverage. Additionally, doing a meaningful cost effectiveness analysis even for the four countries directly modelled in the study, let alone all the other countries to which our results likely apply, would be a program of research unto itself. That said, we appreciate the reviewer's point. To get at a useful comparison in the utility of vaccinating or test-and-treating, we calculated the equivalent number of individuals that would need to either be vaccinated or treated under a low (10%) vaccine coverage scenario to half the number of severe cases.

Line 412: *“Our results suggest that regardless of the (test and antiviral) distribution strategy, an effective test-and-treat program in any country requires large testing rates ($\gg 100$ tests/100K/day) that are far beyond testing rates reported globally since 2023.⁹ In turn, increasing vaccination is likely a more viable approach to lower severe cases than implementing large-scale test-and-treat programs. To compare the vaccination coverage and the resource requirements needed for test-and-treat to achieve the same reduction in disease burden, we computed the additional vaccination coverage needed to halve the number of severe cases at different R_e under 10% starting vaccination coverage. We also estimated the equivalent number of tests and antivirals distributed to half the number of severe cases (Table 2). Across all countries and R_e , we estimated that an additional 24%-67% of the population must be vaccinated to reduce the number of severe cases by half without antivirals. Conversely, ~9,000 – 400,000 courses of antivirals per 1,000,000 people would be needed to avert the same number of severe cases by antivirals for one epidemic wave. Furthermore, we estimated that ~200 – 7,000 tests must be performed per 100,000 people per day to support the distribution of those antivirals. While these testing rates were achieved by some high-income countries during the COVID-19 pandemic, no countries are testing at anywhere near these rates in the post-emergency phase, suggesting that vaccination would likely be the more efficient option for reducing severe disease burden.”*

Other comments:

5. Many references haven't been properly compiled (line 69, 149, 156, 163, 217-218, 219, 248, 264, 282, 479).

Fixed.

6. The figures are overall of high quality. I have a couple suggestions that the authors may consider to improve readability and interpretability. In most figures, countries are ordered alphabetically. I would consider ordering them by increasing income level (i.e. Zambia-Brazil-Georgia-Netherlands) to better depict how the results are impacted by increasing income level.

Countries are now ordered by their income level in all figures.

7. For figure 1b, there is no color scale for the heatmaps.

The color scale is now included in Fig. 1b.

8. It did not strike me at first that most figures were depicting the percentage of severe cases / infections averted due to test-and-treat strategies. To increase understanding by a less quantitative audience, I would consider renaming the y-axis by "Percentage of severe cases averted" instead of "Percentage Δ in severe cases".

All Y-axes labels have been changed accordingly.

9. The color scheme in figures 2-3-4 is pretty. However, I found it a bit hard to understand to trend with increasing testing capacities with the current color schemes that better emphasizes the difference between countries than between testing intensities. I'm wondering whether trying to simply use 3 colours (for each of the different testing intensities) would facilitate interpretation.

Color scheme is now based on testing rates to improve interpretability.

10. In most figures, the linetype (e.g. dashed, plain, pointed) used to depict different testing rates is hard to get from the legend.

We have now used colors instead of line type to depict testing rates.

11. Line 142: Should "Figure 2B" rather be "Figure S3B"?

Corrected.

12. I was wondering how to exactly interpret the reproduction number introduced by the authors. The authors mention it is a combination of population immunity, intrinsic transmissibility, and public health interventions. From the methods, it seems that the transmission rate beta is estimated for a given value of the reproduction number to match the expected growth rate in the absence of interventions. Does this mean that this is done in the absence of vaccination in the population? Or is this procedure repeated for any vaccine coverage considered in the scenario? If not, this might be just fine, but I would clarify that this is the expected reproduction number without vaccination and testing when introducing it. This would also make it clearer that these different scenarios are actually associated with different effective reproduction numbers

at the beginning of the epidemic (because of different vaccine coverages and testing rates).

We address the comments here in our response to comment 2 above.

13. Figure S1 does not show that we are more likely to detect a case when the age distribution is older given that the only thing depicted is the probability to detect symptomatic individuals (line 124-125). One way to do this could be to generate the same graphs but looking at the overall probability for an infected individual to be detected (and not conditional on being symptomatic).

We have changed Figure S2 (previously Figure S1) to show the average probability an infected individual would be detected. To clarify, the age demography of the simulated country is only one component that would affect the average probability of detection. For example, under the same vaccination coverage (50%), same R_e (0.9) and testing rate (500 tests/100,000 people/day), the probability of detection in Zambia, Brazil, Georgia and Netherlands are 42%, 46%, 50% and 59% respectively. The corresponding proportions of individuals aged 60 years and above in these countries are 6%, 15%, 35% and 34%. We had assumed that older individuals are more susceptible to infection (24-47% more likely compared to younger adults) and had a higher probability of becoming symptomatic (80-90% as opposed to 55-80% for younger adults, see Table S1) and thereby, sought testing.

We have stated that Figure S2 showed that (line 130) *“the likelihood of detecting an infection ranged between 0.06% and 64.6%, depending on the country simulated, epidemic intensity, vaccination coverage and test availability (Figure S2). Generally, detection is more likely with a larger proportion of over-60y individuals (i.e. the more likely cases will be symptomatic and seek testing), lower reproduction rate R_e , higher vaccination coverage and greater test availability (i.e. any of the aforementioned factors directly or indirectly increases the surplus of tests available for symptomatic individuals)”*.

14. The contextualization of the results obtained in LMICs with regards to the impact of similar campaigns in HICs is valuable. The intervention scenario used for the Netherlands (in terms of vaccine coverage and testing) is however different from the one used for the LMICs. I understand the motivation of using a realistic scenario for the Netherlands, but I find it harder to compare the results and understand whether the observed differences are attributable to the different demographic structures or interventions considered. I am wondering whether also presenting results for Netherlands with the same intervention as for the LMICs (10-90% vaccine coverage, and a 10 to 500 testing rate) could clarify this.

We have now simulated the same scenarios for Netherlands as we did for LMICs and presented the results alongside those for LMICs.

15. If I understood correctly, treatments are distributed to individuals regardless of their vaccination status. Given the strong protection already conferred by vaccination against severe disease, could restricting the distribution of treatment to high-risk unvaccinated individuals largely increase the impact of the intervention?

See response to comment 16 below.

16. At the beginning of the epidemic wave when infection rates are lower, I expect that testing will not saturate under any testing rate regime explored. However, as the epidemic progresses, I would expect that testing and treatment are “saturating” (i.e. the number of tests performed increases at the beginning of the epidemic and then plateaus when reaching the maximum capacities). When reaching this saturation (or maybe even at the beginning of the outbreak when expecting a larger wave), changing the definition of high-risk individuals (by restricting this group) might increase the overall impact. This could be discussed. The definition of high-risk individuals is indeed currently quite broad and the risk of developing a severe form of the disease upon infection is for example way higher in 80 y.o. than in 60 y.o.

Rationing test-and-treat by changing the definition of high-risk individuals could be a solution to mitigate testing saturation during the epidemic peak. However, there are complex ethical concerns underlying healthcare rationing that is beyond the scope of this work (<https://www.ncbi.nlm.nih.gov/pmc/articles/PMC3415127/>). Furthermore, prioritizing those with the greatest risk of severe disease (i.e. older individuals ≥ 80 years of age or high-risk individuals who were either unvaccinated or partially vaccinated) may not necessarily augment the population-level impact of test-and-treat.

For instance, there are only ~10% of individuals who are ≥ 80 years of age in Georgia who were assumed to have 25% change of developing severe disease if symptomatically infected. However, there are ~25% of individuals who are between 60 and 79 years of age with 13-20% change of become a severe case. Assuming that all of them were infected and present symptomatic infection, there would be ~65% more individuals aged 60-79 years than those ≥ 80 years of age who were not prioritized for treatment. Note that we have not accounted for the fact that older individuals have lower relative contact rates which contributes to likelihood of infection. In terms of prioritizing treatment for unvaccinated/partially vaccinated individuals, while we assumed vaccination could provide 70% protection against severe disease in our model, there is

still a 30% risk of severe disease which can still be augmented by having other risk attributes (e.g. old age and/or co-morbidity).

Reviewer #2 (Remarks to the Author):

This modeling study evaluated the effectiveness of COVID-19 test and treat (via antivirals) strategies on reducing severe infections in low- and middle-income countries (LMIC). Using an agent-based model (ABM) framework, the authors demonstrate that significant reductions in severe cases with antivirals are only possible by substantially increasing testing rates in LMIC and/or restricting testing and availability of antivirals to high-risk populations only. This is a well-written and informative paper that may be useful for direction allocation of limited COVID-19 testing and treatment resources in LMIC for current and future health emergencies. However, there have been other notable studies in this field. This current manuscript would benefit from an overview of previous studies and what this paper adds to the existing literature. Additional comments are provided below.

We thank the reviewer for their comments.

1. Introduction: Please describe other notable studies in this area (e.g., references [1-3])1–3 and discuss what this current study adds to the literature.

We have now briefly described these studies in the Introduction as well and stated how our study adds to the literature:

Line 53: “Various studies have estimated ~10% – 40% reduction in severe disease outcomes if antivirals were distributed to 20% – 50% of all symptomatic infected individuals.^{5,7,8} However, none of these studies have accounted for the diagnostic capacity required to identify and treat these cases with antivirals. There have been substantial gaps in COVID-19 testing equity across country income groups throughout the pandemic. Between January 2020 and March 2022, LMICs were only testing at an average of 27 tests/100,000 people/day (tests/100K/day) as compared to >800 tests/100K/day in high-income countries (HICs).⁹ In the post-public health emergency phase of the pandemic, testing rates have dwindled down to less than 10 tests/100K/day and 100 tests/100K/day on average for LMICs and HICs respectively (as of September 2023).⁹ Low testing rates severely underestimate COVID-19 cases,¹⁰ which not only complicate antiviral demand forecasts but also create additional barriers to the effective distribution and use of antivirals.

Here, we used an agent-based model (PATAT)^{11,12} to demonstrate how testing rates and strategies affect the use and impact of antivirals...”

We had also performed in-depth comparisons of our results against all three studies referred by the reviewer, which we described in the Discussion section in our original and current submission:

Line 431: *“There have been other modelling efforts that estimated substantial reductions in disease burden by distributing antivirals to 20% - 50% of symptomatic infected individuals. However, from our analyses, doing so would also require testing rates that are far greater than 500 tests/100,000 people/day. First, Leung et al.⁷ estimated that distributing antivirals to 50% of all symptomatic infected individuals regardless of risk status would only reduce hospitalizations by 10-13% in a population with high vaccination coverage (70-90%). For the Netherlands, we simulated a population with 80% vaccination coverage and large test availability, that included both clinic-based and over-the-counter self-tests, such that at least 50% of all symptomatic individuals were diagnosed. We estimated that 56-59% of severe cases could be averted if only high-risk symptomatic individuals were administered antivirals. When we reconfigured our simulations to now distribute antivirals to 50% of all symptomatic infected individuals, the proportion of severe cases averted lower to only 18% which is more in line with Leung et al.*

Second, Matrajt et al. found that initiating 20% of infected individuals that were >65 years of age on antivirals daily could avert 32-43% of deaths in an Omicron-like wave ($R_e \geq 2$) for an unvaccinated population in LMICs such as Kenya and Mexico.⁵ We had estimated that 31-62% of deaths could be averted at $R_e = 2$ at low (10%) vaccination coverage in LMICs but only if test availability was 500 tests/100K/day and clinic-provided symptomatic testing were restricted to high-risk individuals, which would mean a daily average of 19-20% of high-risk infected individuals being initiated on treatment each day. If there are no restrictions on access to clinic-provided tests, testing rate must be at least 750 tests/100K/day to initiate 20% of infected >65-years on antivirals daily with >95% probability, indicating that the previous from Martrajt et al. predicated on very high testing rates.

Finally, Brault et al. estimated that 11% of hospitalizations could be averted if antivirals with 50% effectiveness were administered to half of all high-risk cases in Wallis and Futuna, where ~70% of individuals have at least two doses of vaccines, during an epidemic wave with a doubling time of 2-3 days.⁸ In the closest scenario we had simulated (i.e. $R_e = 2$, 46% effectiveness of antivirals, 50% vaccination coverage and 500 tests/100K/day), we estimated that severe cases could be reduced by 7% in Brazil (Figure 2B), which has a similar demography to Wallis and Futuna (i.e. median age = 33 and 35 years in Brazil and Wallis and Futuna respectively; proportion of individuals ≥ 65 years = 10% and 13% respectively). However, like the two preceding examples, this is only

possible at testing rates that are many-fold higher than those performed in most LMICs both during and after the emergency-phase of the pandemic.”

2. Methods: More detail needed on modeling the impact of antivirals on transmission reduction

Clinical trials of Paxlovid did not report significant reductions in onward transmissions between household contacts. As such, we did not model the direct impact of antivirals on transmission reduction in our simulations. However, as described in the Methods, individuals at risk of developing severe disease who were effectively treated with antivirals will experience mild disease with a shorter recovery period, and thus infectious period than if they were to present severe symptoms. There could in turn result in indirect impacts on reducing onward transmission. Nonetheless, based on our simulations, these indirect impacts are limited in their ability to reduce total infections because (line 142) *“58-67% of all transmission events were attributed to asymptomatic and pre-symptomatic individuals (Figure S4A)”*.

We have now clarified the further in the Methods (line 674):

“We did not factor any risk reduction in transmissions or deaths given the lack and low certainty of evidence of the impact of oral antivirals on protection against transmission and mortality respectively.⁴⁷ However, individuals could only die from COVID-19 if they had progressed to severe disease in our model.

For individuals who were treated with antivirals that were simulated to result in severe disease, we performed a Bernoulli trial with the probability of averting severe disease (i.e. 46%), provided that they were currently in the presymptomatic phase or were experiencing mild disease. If the Bernoulli trial succeeded, we re-simulated their disease progression and within-host viral dynamics using the procedures above but now under the assumption that they would develop only mild disease and conditioning that the maximum viral load is lower than before. Changes were only made to the upcoming phases of disease progression from the then current phase of infection. The average recovery period of having mild symptoms was assumed to be 5.4 days as opposed to 18.1 days when presenting severe disease.^{41,48} In turn, while we did not parameterize the impact on transmission reduction by antivirals, the shortened recovery, and thus infectious period as well as lower maximum viral load of individuals who were effectively treated could result indirect reduction in onward transmissions.”

3. Results (lines 83-89) and Discussion (lines 460-464): What are the potential consequences of consolidating the effects of immunity from previous infection into

instantaneous reproduction number (R_t)? Is this expected to yield bias or reduce precision?

See response to reviewer 1, comment 2.

4. Results: The finding that increasing vaccination coverage does not change the impact of antiviral distribution on infections averted appears counterintuitive given that vaccines are highly protective against severe infection. Can the authors provide some rationale for this finding?

We have now more clearly explained how vaccination coverage changes the impact of antiviral distribution. See response to reviewer 1, comment 2.

5. Results. While 2-dose mRNA vaccines are less protective against general infection, studies have found that booster doses are protective against general infection from omicron.⁴ Were booster doses considered in this study? The authors may consider tying this into the sensitivity analyses results on lines 310-323.

Yes, we had considered reported vaccine effectiveness for different vaccines and doses, including booster doses – line 100: *“For comparability between countries and as a simplification, we assumed that protection rates against infection and severe disease were 29% and 70% respectively, which were based on the more conservative, lower average estimates of vaccine effectiveness against BA.1 across different vaccines (i.e. mRNA and ChAdOx1 nCoV-19 vaccine) and doses (i.e. 1-3 doses).¹⁹⁻²¹”*

In our sensitivity analyses, we have already considered scenarios with even more protective vaccines (line 332, either *“52% and 96% protection against infection and severe disease respectively”* or *“75% and 97% protection against infection and severe disease respectively”*).

Line 328: *“Second, we had assumed low average estimates of vaccine effectiveness (i.e. 29% and 70% protection against infection and severe disease respectively). However, vaccine effectiveness can be improved by updating the vaccine strains to match circulating viruses or through booster shots. We repeated our simulations with vaccines conferring greater effectiveness, including known average protection against Delta-like (i.e. 52% and 96% protection against infection and severe disease respectively) and wild-type SARS-CoV-2 viruses (i.e. 75% and 97% protection against infection and severe disease).¹⁹⁻²¹ Similar to our original results for low vaccine effectiveness, use of antivirals could reduce transmissions in Georgia by up to ~20% but only if testing rates were high (500 tests/100K/day; Figure S16). In contrast, the proportion of severe cases averted due to antivirals became increasingly uncertain (i.e. wider error bars in Figure S17). This was*

because improved vaccine effectiveness, on top of wider vaccination coverage, substantially reduced the number of severe cases. Nonetheless, regardless of vaccine effectiveness and coverage, meaningful reductions in severe cases by antivirals could only be achieved with higher testing rates (≥ 100 tests/100K/day) to support the administration of antivirals for infected high-risk individuals."

6. Results (lines 265-266). Several studies have demonstrated that vaccines are protective against severe SARS-CoV-2 infection.⁵ If antivirals were to be prioritized, should vaccination status not be accounted for?

See response to reviewer 1, comment 15.

7. Discussion. In this section, it would be helpful to link different values of the instantaneous reproduction number (R_t) to known events. For example, on line 361-362 the authors could phrase this as "..., could average severe cases by up to 65% in LMICs experiencing an epidemic wave that initialized at $R_t \leq 1.2$ (e.g., BA.5 variant-like event).

We have now contextualized any R_e values mentioned in the Discussion section.

8. Discussion (lines 377-380). Another potential strategy is simply testing all high-risk individuals in a household if any individual within that household is infected. Previous research has showed such strategies have a substantially higher probability of detecting infected cases.⁶

Indeed, and we had already done so in the original submission: see Results, line 187 (*Distribution of test and antivirals to high-risk household contacts of test-positive individuals*) and Discussion, line 390: The reason why this strategy yielded lower reductions in severe cases ("two to ten-fold" lower) is because "*setting aside tests to screen high-risk household contacts under test availability constraints diminish the utility of tests that would have otherwise been used to test symptomatic individuals who sought testing*".

We have also further commented on another potential strategy of distributing antivirals to high-risk household contacts of test-positive individuals without the need for testing themselves:

Line 396: "*A potential workaround could be to distribute antivirals to high-risk household contacts of test-positive individuals without the need to confirm if the high-risk contacts were infected themselves by testing. However, this would also increase the number of antiviral courses needed as well as result in wastage among individuals who were not*

infected. A cost effectiveness analysis could be performed to identify the most resource effective strategy but is beyond the scope of this work.”

9. Discussion (lines 433-436). I suggest the authors perform a sensitivity analysis to the assumption that limits timely access to antivirals, as this is certainly a barrier in many LMIC.

We agree. We have showed that, even under a high testing rate scenario for the Netherlands, <20% of treated high-risk individuals would have received their antivirals in a timely manner (i.e. within five days since symptom onset) if there were >2 days of delay in treatment administration. These results should apply to LMICs as well. We have now discussed this in line 482:

“Third, we had assumed that clinical testing for disease and administration of treatment occur on the same day in our simulations. However, any practical barriers that limit timely access to antivirals (e.g. inadequate supply and distribution, limited access to healthcare providers, acceptance of antiviral therapy) can substantially reduce the estimated impact of test-and-treat programs.³⁵ As shown in Figure S12, even under a large test availability scenario (with self-tests), if administration of antivirals was delayed by >2 days, <20% of high-risk treated individuals received their antiviral courses within the 5 days post-symptom onset window when Paxlovid was reported to be efficacious. As such, even if testing rates could sufficiently support test-and-treat programs, delays in accessing antivirals, which had been reported in various LMICs,³⁶ must be minimized for these programs to remain effective. Ideally, testing and treatment of infected patients should occur at the same clinical interaction.”

10. Introduction/Discussion. There are currently strategies in place for prioritizing administration of Covid-19 treatment (e.g., Paxlovid; <https://www.cdc.gov/coronavirus/2019-ncov/hcp/clinical-care/outpatient-treatment-overview.html>). However, strategies that have not been implemented are not discussed in the context of this study.

According to the US CDC guidelines (as stated in the website cited by the reviewer), antivirals “are the preferred treatments for eligible adult and pediatric patients who are at high risk for progression to severe COVID-19. Clinicians should consider COVID-19 treatment in patients with mild-to-moderate COVID-19 who have one or more risk factors for severe COVID-19. Treatment must be started as soon as possible and within 5 days of symptom onset to be effective.” Risk factors for severe COVID-19, according to the same guidelines, include: “(1) Age over 50 years, with risk increasing substantially at age \geq 65 years; (2) Being unvaccinated or not being up to date on COVID-19

vaccinations; (3) Specific medical conditions and behaviors” which include individuals with relevant co-morbidities.

We have implemented a nearly identical strategy as described above and more:

Line 108: *“We assumed that only high-risk individuals (i.e. ≥ 60 years of age (over-60y) or an adult ≥ 18 years with a relevant comorbidity) who tested positive at clinics (e.g. a self-reported self-test would be insufficient to access antivirals) would receive a course of antivirals.”*

Line 669: *“Regardless of their vaccination status (per WHO guidance),⁴⁷ all high-risk individuals who tested positive within five days after symptom onset were eligible for a course of antiviral therapy.^{3,4}”*

11. Methods (diagnostic testing, lines 589-590). Please provide more detail on how this was estimated, including the role of access ability to healthcare facilities.

We have now provided further details in the Methods.

Line 632: *“Based on the range of test positivity rates reported by various countries during the second-half of 2021 (when community testing was assumed to still be prevalent in most countries),⁴⁵ we assumed that test positivity rate was 10% at the start as well as end of an epidemic wave, and a 20% test positivity rate at the peak, linearly interpolating the demand for periods between these time points.^{11,12}”*

We have now clarified that we assumed that there is an (line 653) *“average 65% chance that an individual with mild symptom would seek clinic-provided testing and were only tested if there were available test stocks”*.

12. References: There are several instances of references not working/linking.

Fixed.

13. Appendix (household, lines 33-34). Please provide more explanation and detail on how households were ordered to implicitly approximate neighborhood proximity.

We have now provided more explanation as to how households were ordered to approximate neighbourhood proximity (Supplementary Information):

“Although PATAT does not explicitly model the geolocation of agents, households are ordered to implicitly approximate neighbourhood proximity. Herein, households are

assigned with a numerical identifier. The smaller the difference between the household numerical identifiers, the nearer the households were assumed to be by location.”

14. Appendix (model validation, lines 80-92). Why was the ABM only validated in one country prior to March of 2021? Given there have been several high-profile variants in the last 2 years (including Delta and Omicron variants), the authors should consider validating this model in more relevant settings.

See response to reviewer 3, comment 1.

15. Appendix (Table S1). In addition to providing distributional assumptions for input parameters, the authors may consider providing the median (IQR) or mean (SD) of these distributions. This will be more understandable to lay audiences and makes it easier to evaluate underlying assumptions.

We have now explicitly stated which values in Table S1 are the mean and SD of these distributions.

References

1. L M, Er B, Ms C, D D, H J. Could widespread use of antiviral treatment curb the COVID-19 pandemic? A modeling study. BMC infectious diseases. 2022;22(1). doi:10.1186/s12879-022-07639-1
2. Leung K, Jit M, Leung GM, Wu JT. The allocation of COVID-19 vaccines and antivirals against emerging SARS-CoV-2 variants of concern in East Asia and Pacific region: A modelling study. Lancet Reg Health West Pac. 2022;21:100389. doi:10.1016/j.lanwpc.2022.100389
3. Brault A, Tran-Kiem C, Couteaux C, et al. Modelling the end of a Zero-COVID strategy using nirmatrelvir/ritonavir, vaccination and NPIs in Wallis and Futuna. Lancet Reg Health West Pac. 2023;30:100634. doi:10.1016/j.lanwpc.2022.100634
4. Rennert L, Ma Z, McMahan C, Dean D. Covid-19 vaccine effectiveness against general SARS-CoV-2 infection from the omicron variant: A retrospective cohort study. PLoS Global Public Health. 2023;3(1):e0001111. doi:https://doi.org/10.1371/journal.pgph.0001111
5. Bloomfield LE, Ngeh S, Cadby G, Hutcheon K, Effler PV. SARS-CoV-2 Vaccine Effectiveness against Omicron Variant in Infection-Naive Population, Australia, 2022 - Volume 29, Number 6—June 2023 - Emerging Infectious Diseases journal - CDC. doi:10.3201/eid2906.230130
6. Rennert L, McMahan C, Kalbaugh CA, et al. Surveillance-based informative testing for detection and containment of SARS-CoV-2 outbreaks on a public university campus: an observational and modelling study. The Lancet Child & Adolescent Health. 2021;5(6):428-436. doi:10.1016/S2352-4642(21)00060-2

Reviewer #3 (Remarks to the Author):

This is a very well-presented and comprehensive study detailing the assessment of the impact of a test-to-treat program with rapid tests and antivirals on SARS-CoV-2 infection outcomes. The analyses are largely appropriate, with considerable attention being paid to determining the sensitivity of the results to variations in key parameters.

We thank the reviewer for the helpful comments.

1. However, the analysis focuses primarily on scenarios in which a relatively small proportion of the population initially infected (1% and lower in sensitivity analyses) with a 90-day epidemic wave subsequently following, as may have occurred during the pandemic phase; population immunity (beyond age-specific susceptibility and vaccine-derived immunity) is not accounted for in the model due to complexities in individual infection and vaccination histories between countries and over time. Given it appears that we are in the post-crisis, “endemic” phase, it would be prudent to assess test-to-treat in additional scenarios with a higher prevalence of infection, varying between 1-10%, but with R averaging 1 over a longer period (i.e., as observed by the UK ONS Covid Infection Survey in 2022-2023 with the Omicron subvariant waves <https://www.ons.gov.uk/peoplepopulationandcommunity/healthandsocialcare/conditionsanddiseases/bulletins/coronaviruscovid19infectionsurveypilot/24march2023>), and low severity of infection (due to population immunity). It is not clear how future SARS-CoV-2 dynamics will proceed given the waning of infection and vaccine-derived immunity as well as the emergence of variants, though a high-prevalence endemic state with lower severity is possible and hence test-to-treat should be assessed in this context. This should also be talked about in the discussion.

We have, in fact, accounted for a range of effective reproduction numbers (i.e. $R_e=0.9, 1.2, 1.5$ and 2.0) that coincide with the spread of more recent “endemic” Omicron subvariants (e.g. BA.5 and XBB.1.5). See line 83: *“We simulated SARS-CoV-2 epidemics in each country under a range of average effective reproduction number (i.e. $R_e = 0.9, 1.2$ (doubling time = 6-9 days), 1.5 (doubling time = 3-5 days), and 2.0 (doubling time = 1-3 days)) during the first week of each simulation. These doubling times coincide the range reported for prominent Omicron subvariants as well, including BA.2 (~3 days)¹⁶, BA.5 (5-6 days)¹⁷ and XBB.1.5 (9-10 days)¹⁸ (Figure S1). All simulations were initialized with 1% of the population infected at the start of the epidemic.”*

To be clear, the simulated epidemics were initialized at 1% prevalence (i.e. 1% of the population actively infected) at the start of the epidemic. Prevalence would change over time as the virus spread across the population. As additional validation, we used the UK OBS Covid Infection Survey result cited by the reviewer and showed that the prevalence estimates of our simulations lie within the reported range in the UK for BA.5 and XBB.1.5:

Supplementary Information: “To validate our model, we used the estimated percentage population that would test positive for COVID-19 infection, regardless of whether they reported that they were experiencing symptoms, during the spread of Omicron BA.5 (effective reproduction number (R_e) = ~ 1.5) and XBB.1.5 (R_e = ~ 1.2) subvariants in the UK, including England, Wales and Scotland

(<https://www.ons.gov.uk/peoplepopulationandcommunity/healthandsocialcare/conditionsanddiseases/bulletins/coronaviruscovid19infectionsurveyypilot/24march2023>). These

estimates reflect the community prevalence of SARS-CoV-2 in the UK based on testing results from random community supervised self-swabbing RT-PCR-based surveillance collected across the country¹ ($\sim 300,000$ swab tests per month;

<https://www.ons.gov.uk/peoplepopulationandcommunity/healthandsocialcare/conditionsanddiseases/methodologies/coronaviruscovid19infectionsurveygmi> for more details).

For the same variant R_e values, the UK estimates fitted well within the 95% bootstrap confidence interval of the proportion of infectious individuals, regardless if they presented symptoms, that would test positive over time in our PATAT simulations.”

Figure S1: Model validation – Comparing simulated prevalence against estimated UK prevalence. The blue line shows the average simulated percentage of the population that would test positive for COVID-19 each day (i.e. percentage of individuals that were productively infectious, regardless if they were symptomatic, and would test positive for SARS-CoV-2) for different effective reproduction number (R_e). The blue shaded area shows the range of simulated percentages across all simulations for all four simulated countries (i.e. Zambia, Brazil, Georgia and the Netherlands), under all testing rates and vaccination coverage, with no test-and-treat programs. The scatter points in subplots of $R_e = 1.2$ and $R_e = 1.5$ show the estimated community prevalence data from different countries (i.e. England, Wales and Scotland) in the UK

(<https://www.ons.gov.uk/peoplepopulationandcommunity/healthandsocialcare/conditionsanddiseases/bulletins/coronaviruscovid19infectionsurveyypilot/24march2023>) during

the spread of BA.5 ($R_e \sim 1.5$) and XBB.1.5 ($R_e \sim 1.2$). The UK prevalence estimates were based on testing results from random community supervised self-swabbing RT-PCR-based surveillance collected across the country¹ ($\sim 300,000$ swab tests per month;

<https://www.ons.gov.uk/peoplepopulationandcommunity/healthandsocialcare/conditionsanddiseases/methodologies/coronaviruscovid19infectionsurveyqmi> for more details).

We have now also stated, as a limitation of our work, that it is unclear how future immunity waning dynamics could impact the level of severe disease outcomes in the population (line 504):

“Finally, we did not factor in changes to individual immunity levels due to previous infections or immune waning. As a simplification, we assumed that these effects have been implicitly captured by various initial R_e values and were able to simulate epidemics with prevalence ranges similar to those reported during the spread of Omicron subvariants BA.5 and XBB.1.5 (Figure S1). However, it is currently unclear how changing immunity dynamics in the future could affect severe disease outcomes.”

2. Methods and supplement: Unless I have missed it, it is not stated how vaccines protect against death (infection and severe disease are noted on line 93; it would be good to have these in the table). This should be made explicit as it is a key parameter in the model.

The vaccine effectiveness estimates we assumed were based on those against severe disease outcomes which were defined as hospitalization and/or death (Andrews et al., 2022, Buchan et al., 2022). Only Tseng et al., 2022 estimated for vaccine effectiveness against hospitalization but reported comparable effectiveness estimates with Andrews et al and Buchan et al. As we were unable to distinguish the specific effectiveness against death, likewise with antivirals, we assumed vaccine effectiveness estimates as the protection against severe disease which would implicitly protect against deaths as well since individuals could only die from COVID-19 if they had progressed to severe disease in our model. We have now made this clear in the Methods.

Line 622: *“We did not assume a specific protection rate against death since the referenced studies had reported effectiveness estimates against severe disease outcomes which include hospitalization and/or death. Nonetheless, protection of deaths is implicitly accounted for since individuals could only die from COVID-19 if they had progressed to severe disease in the model.”*

Additional comments:

3. There are several missing references in the text (Error! Reference source not found.)

Fixed.

4. Line 486: dates should be referenced clearly (what does “last two years” of the pandemic refer to?) throughout the text.

We have revised the main text to refer all mentioned time periods by specific dates.

REVIEWERS' COMMENTS

Reviewer #1 (Remarks to the Author):

The authors have appropriately addressed all my concerns. The minor modifications made to the figures really do increase readability and interpretability, thank you!

I just have two minor points:

- Table 1 (the inclusion of absolute case reduction is really helpful!) has some typos I believe (a lot of 0).
- It would be nice to include a reference to this new table in the text when it is referred to (paragraph starting at line 177).
- Line 388: seek should be seeking.

Congratulations on a nice paper, I don't have any additional comments and recommend the manuscript for publication.

Reviewers' comments in blue; Authors' response in black.

Reviewer #1 (Remarks to the Author):

The authors have appropriately addressed all my concerns. The minor modifications made to the figures really do increase readability and interpretability, thank you!

We thank the reviewer for the constructive comments throughout the review process.

I just have two minor points:

- Table 1 (the inclusion of absolute case reduction is really helpful!) has some typos I believe (a lot of 0).

All typos have been corrected.

- It would be nice to include a reference to this new table in the text when it is referred to (paragraph starting at line 177).

Reference to table 1 is now included at line 175.

- Line 388: seek should be seeking.

Changed.